# Elastomeric polyamide biomaterials with stereochemically tuneable mechanical properties and shape memory

Joshua C. Worch [1], Andrew C. Weems[1], Jiayi Yu [2], Maria C. Arno [1], Thomas R. Wilks [1], Robert T. R. Huckstepp [3], Rachel K. O'Reilly [1], Matthew L. Becker [4✉] & Andrew P. Dove [1✉]

Biocompatible polymers are widely used in tissue engineering and biomedical device applications. However, few biomaterials are suitable for use as long-term implants and these examples usually possess limited property scope, can be difficult to process, and are non-responsive to external stimuli. Here, we report a class of easily processable polyamides with stereocontrolled mechanical properties and high-fidelity shape memory behaviour. We synthesise these materials using the efficient nucleophilic thiol-yne reaction between a dipropiolamide and dithiol to yield an $\alpha,\beta$ — unsaturated carbonyl moiety along the polymer backbone. By rationally exploiting reaction conditions, the alkene stereochemistry is modulated between 35–82% *cis* content and the stereochemistry dictates the bulk material properties such as tensile strength, modulus, and glass transition. Further access to materials possessing a broader range of thermal and mechanical properties is accomplished by polymerising a variety of commercially available dithiols with the dipropiolamide monomer.

---

[1] School of Chemistry, University of Birmingham, Edgbaston, Birmingham B15 2TT, UK. [2] Department of Polymer Science, The University of Akron, Akron, OH 44325, USA. [3] School of Life Sciences, University of Warwick, Coventry CV4 7AL, UK. [4] Department of Chemistry, Department of Mechanical Engineering & Materials Science, Department of Orthopaedic Surgery, Duke University, 308 Research Drive, Durham, NC 27708, USA. ✉email: matthew.l.becker@duke.edu; a.dove@bham.ac.uk

The ability to rationally modulate the thermomechanical properties of polymeric materials by design is a fundamental aim of materials science. It is well known that stereochemistry of polymers dictates their bulk properties, but the importance of stereocontrol in polymer synthesis is often overlooked. For example, the naturally occurring isomers of polyisoprene—natural rubber and gutta-percha—display stereochemically dependent thermomechanical properties where the *cis* orientation of the alkene moiety disrupts chain packing and leads to a much softer, more amorphous material[1,2]. Such striking structure–property relationships are often found in stereochemically precise biopolymers, e.g., collagen or elastin, but there remain significant limitations in synthetically mimicking these complex biological materials[3,4]. Achieving control over stereochemical assembly of monomers into synthetic polymers, particularly backbone stereochemistry, could afford a simple platform from which to access a diverse range of materials properties. This is of particular importance when considering the unique mechanical needs of biomaterials operating in diverse physiological environments[5,6].

Synthetic polymers have been used in medical devices for more than five decades, and key recent advances have largely focused on the development of biodegradable (or resorbable) materials for tissue engineering[7]. As such, the continued innovation of long-lasting non-resorbable polymers in joint and/or bone therapies, for example, are lagging behind and suffer from notable limitations such as wear, difficult processing, sterilisation or high cost. Nevertheless, polyamides have been a biomaterial of choice for decades and they have been extensively developed for applications ranging from use as sutures[8,9] or membranes[10] to vascular applications[11] because of their toughness, low cost and outstanding biocompatibility[12]. Polyamides are also widely used in bone engineering as a consequence of the materials' high strength and flexibility, which is due to the extensive degrees of both crystallinity and hydrogen bonding[13–21]. Even though this may be useful in selected orthopaedic or vascular applications, these types of materials are typically difficult to process and functionalise, which has ultimately limited their performance in other applications. An ideal durable biomaterial platform would incorporate the thermomechanical and biological performance of polyamides while displaying enhanced processability and advanced functionality, such as shape memory, for minimally invasive device designs.

We set out to create a tough, resilient biomaterial with stereochemically dependent properties by combining a polyamide backbone with unsaturated alkene moieties. Synthetic heteroatom-containing polymers featuring stereocontrolled unsaturated moieties along the main chain are not very abundant. Nonetheless, polyesters or poly(ester-amide)s[22–24], prepared with pre-defined stereochemical units, such as fumaric acid, have garnered attention as biomaterials[25–34]. However, maleate-based materials have been notoriously hard to prepare due to isomerization issues[35,36] and, with the exception of a recent report[37], only low-molecular-weight polymers have been isolated[38,39], which reduces the durability of the materials. Furthermore, stereochemistry has not been directly invoked in order to modulate properties in most of these examples. Synthetic polyamides possessing chiral centres are comparatively abundant and well represented in the literature with the stereochemistry usually derived from the use of naturally occurring comonomers and/or precursors, such as sugars[40–48], amino acids[49,50], tartaric acid[51–53] and terpenes[54–56]. A central feature among the stereo-defined polyamides is their inherent crystallinity, which has been assessed in most instances. However, polymer mechanical properties and how those relate to stereochemistry are generally absent from these studies, presumably due to the challenges in synthesising high-molecular-weight step-growth polymers without implementing harsh reaction conditions.

Herein, we explore the mild nucleophilic thiol-yne addition of dithiols to a dipropiolamide monomer to produce stereocontrolled unsaturated polyamides possessing robust thermomechanical properties. The presence of both amide and unsaturated moieties along the backbone imbue the materials with unique properties, namely shape-memory behaviour and stereochemically dependent thermomechanical properties. Moreover, the thiol-yne polyamides are amorphous, with good optical transparency, and also exhibit elasticity at temperatures above the glass transition. These properties make the polymers unique among synthetic polyamides, such as Nylons, which are semi-crystalline (when stretched) and display no shape-memory behaviour, factors which limit their utility as the medical device community begins to embrace minimally invasive designs. All materials are cytocompatible and preliminary in vivo experiments indicate excellent biocompatibility, underscoring their potential as versatile biomaterials.

## Results

**Synthesis of stereocontrolled polyamides.** Recently, the nucleophilic Michael addition reaction between nucleophiles and activated alkynes (amino-yne, phenol-yne and thiol-yne) has attracted considerable attention as an efficient path to step-growth polyesters[57,58]. There has been some progress in controlling the stereochemistry of the resultant alkene in the backbone[59–65], but many "yne-type" polymers are not synthesised in a stereocontrolled manner, despite the well-established impact of double-bond stereochemistry on bulk material properties (e.g., glass transition and stiffness)[60]. Initially, we screened reaction conditions that were used for the analogous stereocontrolled reaction between thiols and propiolate esters[63]. The combination of a dipropiolamide monomer ($C_3A$) (Supplementary Figs. 1 and 2), 1,6-hexanedithiol ($C_6T$) and 1 mol% 1,8-diazabicyclo[5.4.0] undec-7-ene (DBU) in dimethyl sulfoxide (DMSO) afforded a fibrous solid after isolation from the reaction mixture (Fig. 1a). After drying, the solid was analysed using $^1H$ NMR spectroscopy and the product showed characteristic vinyl proton resonances (73% *cis*) indicative of a thiol-yne addition (Fig. 1b; Supplementary Figs. 3 and 4). Finally, size-exclusion chromatography (SEC) confirmed high molar mass polymers (Fig. 1c; Supplementary Fig. 20).

Next, the stereochemistry of the vinyl thioether moiety in the polymer backbone was modified by changing either the polarity of the reaction medium and/or the base catalyst[60]. Although DMSO was critical for polymer solubility, the overall polarity of the reaction mixture could be modulated to be less polar (dilution with chloroform) or more polar (dilution with methanol). Qualitatively, the reaction between the propiolamide and thiol is sluggish compared with the propiolate analogue[60], thus limiting the choice of suitable base catalysts. A variety of organic bases were screened, including triethylamine (NEt₃), only yielding oligomeric species. However, by employing 1,4-diazabicyclo[2.2.2]octane (DABCO) at higher loadings (10 mol%), we could isolate high molar mass polymers ($M_w \geq$ 100 kDa) and we chose DABCO as the "weak" base (p$Ka$ (H-DABCO) = 8.8 versus, p$Ka$ (H-DBU) = *ca*. 12). Our initial reactions (using DBU and DMSO) afforded polymers with moderately high *cis* content (*ca*. 70%) (Table 1, entry P2). The *cis* content could be increased by adding methanol to the DMSO/DBU reaction mixture (Table 1, entry P1; Supplementary Figs. 5 and 6). To decrease the *cis* content in the resultant polymers, progressively greater amounts of chloroform were added to the DMSO reaction mixture, and DBU was

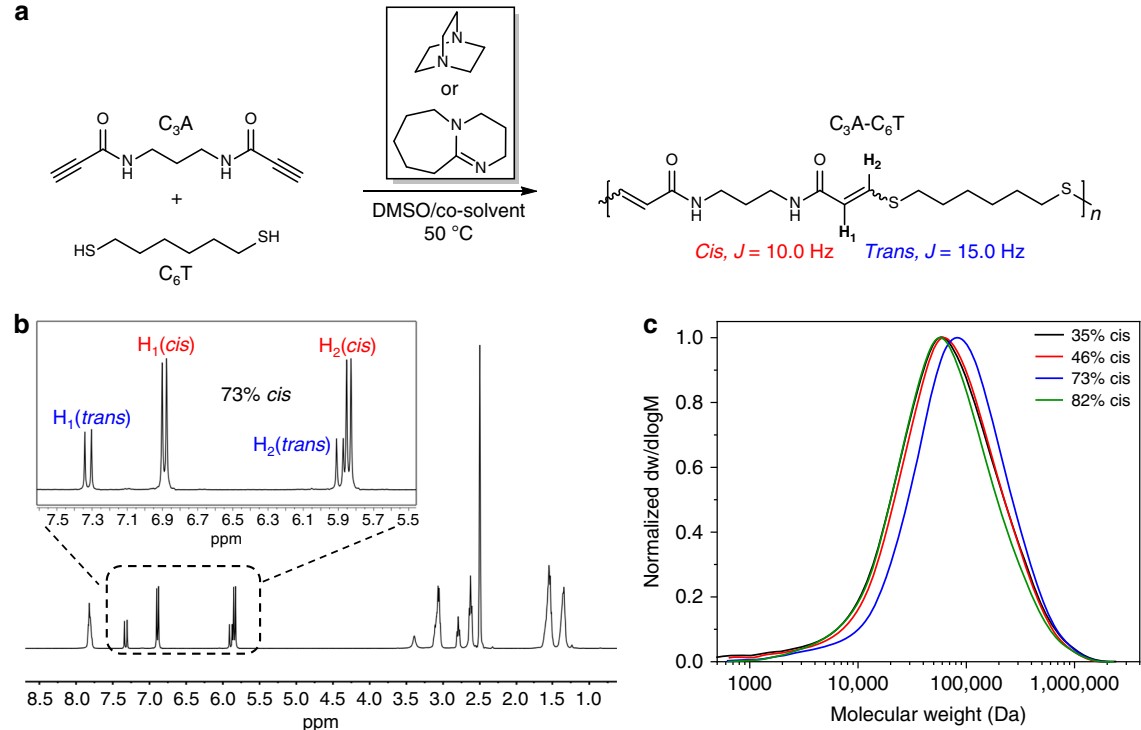

**Fig. 1 Characterisation of stereocontrolled polyamides. a** Step-growth polyaddition reaction between $C_3A$ and $C_6T$ to furnish unsaturated polyamides ($C_3A$-$C_6T$). **b** Representative $^1H$ NMR spectrum (DMSO–$d_6$, 25 °C, 400 MHz) of $C_3A$-$C_6T$ (73% *cis* content) from using DMSO and DBU. *Cis* and *trans* proton resonances were assigned by coupling constants. **c** SEC (DMF, 0.5 % w/w $NH_4BF_4$) chromatograms determined against poly(methyl methacrylate) (PMMA) standards for polyamides with various stereochemistry.

**Table 1 Characterisation of material properties for polymers obtained using different polymerisation conditions for $C_3A$ and $C_6T$ using 1 mol% base at [$C_3A$] = 1 M.**

| Entry | Solvent | Base | %cis[a] | $M_w$ (kg mol$^{-1}$)[b] | $E$ (MPa) | $\varepsilon_{break}$ (%) | UTS (MPa) | $T_g$ (°C)[c] |
|---|---|---|---|---|---|---|---|---|
| P1 | DMSO/MeOH (2:1) | DBU | 82 | 104.6 | 1278.4 ± 42.1 | 109 ± 28 | 70.4 ± 2.6 | 98 |
| P2 | DMSO | DBU | 73 | 131.4 | 1219.3 ± 48.0 | 158 ± 55 | 67.9 ± 4.2 | 94 |
| P3 | DMSO/CHCl₃ (1:1) | DBU | 46 | 111.8 | 1111.7 ± 19.6 | 158 ± 45 | 60.9 ± 3.9 | 83 |
| P4 | DMSO/CHCl₃ (1:2) | DABCO[d] | 35 | 112.5 | 1052.5 ± 74.2 | 138 ± 87 | 65.5 ± 8.3 | 84 |
| Nylon 6 | | Commercial sample | | | 229.6 ± 53.6 | 337 ± 59 | 67.1 ± 3.9 | 45 |
| Nylon 6,6 | | Commercial sample | | | 1249.3 ± 64.3 | 441 ± 11 | 87.1 ± 5.4 | 65 |

[a]%*cis* content determined by $^1H$ NMR spectroscopic analysis.
[b]$M_w$ determined by SEC (DMF, 0.5% w/w $NH_4BF_4$) analysis against poly(methyl methacrylate) (PMMA) standards. Mechanical data (E, ε-break and UTS) calculated from $n \geq 3$ samples. ± represents 1 s.d. Nylon 6,6 was averaged from $n = 2$ samples. Nylon 6 and Nylon 6,6 were commercial samples and are not soluble in the reported SEC solvent.
[c]Reported from the DSC thermograms of 2nd heating cycle.
[d]In total, 10 mol% (relative to monomer) was used.

substituted with DABCO (Table 1, entries P3–4; Supplementary Figs. 7–10).

Additional propiolamide monomers with longer methylene spacers (such as $C_4A$, $C_5A$ and $C_6A$) were active in the step-growth polyaddition reaction using our optimised conditions, but the resultant material would precipitate from the reaction mixture as an intractable powder. Since we could not adequately characterise these polyamides because of their low processability, we excluded them from this study.

**Thermal and mechanical properties**. The thermal properties of the materials were initially probed using thermogravimetric analysis (TGA) and the degradation temperatures ($T_d$) were

≥300 °C, indicating excellent thermal stability (Supplementary Fig. S21). Differential scanning calorimetry (DSC) of the polymers showed a broad glass transition ($T_g$) and no discernible melt transition ($T_m$) (Fig. 2a; Supplementary Fig. 22). Nevertheless, the $T_g$ displayed a positive correlation with %*cis* content as indicated by $\Delta T_g = 14$ °C between P4 (35% *cis*) and P1 (82% *cis*) (Fig. 2b; Supplementary Fig. 23). The thermal behaviour suggested that the materials were largely amorphous and surface imaging of P1 using atomic force microscopy (AFM) also revealed a non-crystalline morphology (Supplementary Fig. 39). Prolonged annealing below and/or above the respective glass transition temperature also did not induce any crystallisation in P1 (Supplementary Fig. 26). Repeated cycling of the polymers during thermal analysis also led to a decrease in modulus, which may

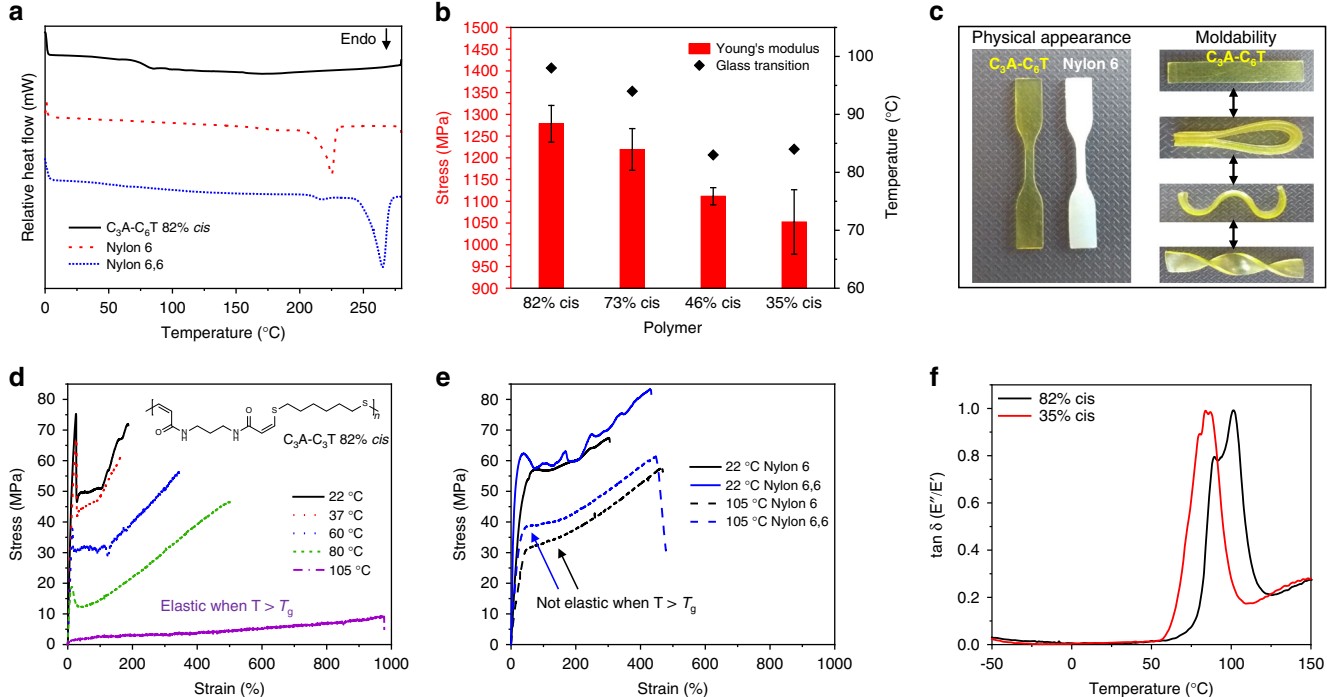

**Fig. 2 Thermomechanical characterisation of the materials. a** DSC thermograms of 1st heating cycle for $C_3A$-$C_6T$, Nylon 6 and Nylon 6,6 from 0 – 280 °C, 10 °C min$^{-1}$. **b** Bar chart with thermomechanical properties for $C_3A$-$C_6T$ with various *cis* content. Error bars represent 1 s.d. **c** Physical appearance/ moldability of $C_3A$-$C_6T$ and Nylon 6. **d** Representative stress vs strain curves for $C_3A$-$C_6T$ ($T_g = 98$ °C) at various temperatures tested at 10 mm min$^{-1}$. **e** Representative stress vs strain curves for Nylon 6 and Nylon 6,6 ($T_g$s < 105 °C) at various temperatures tested at 10 mm min$^{-1}$. **f** DMA temperature sweep comparing $C_3A$-$C_6T$ (high *cis* and high *trans*).

have been due to undesirable cross-linking reactions or possibly isomerization of the alkene moiety. However, according to $^1$H NMR spectroscopic analysis, negligible isomerization of the alkene (73 versus 71% *cis*) occurred after protracted heating of a sample (*ca.* 12 h, 150 °C, N$_2$ atmosphere) (Supplementary Fig. 19).

It became clear that the properties of the unsaturated polyamides were very different from those of common saturated polyamide analogues, such as Nylons, and it was perhaps most obvious when comparing their physical appearance (Fig. 2c). Variable temperature tensile testing of the high *cis* polymer showed that as the polymer approached its glass transition temperature it softened significantly, with a concomitant improvement in ductility (Fig. 2d). To our surprise, when testing the material above the glass transition temperature we observed elastomeric behaviour; this occurrence contrasts distinctly with the thermoplastic nature of Nylons under similar conditions (Fig. 2e). However, the mechanical profiles of both classes of polyamides were comparable at ambient temperature (Supplementary Fig. 33). These results suggest that this class of stereocontrolled, unsaturated polyamides can match the mechanical strength of Nylons, but with the added benefit of temperature-tuneable mechanical properties. As a control experiment, we synthesised a Nylon 3,14 derivative in order to better model the composition of the $C_3A$-$C_6T$ polymers. As expected, this polymer was similar to the commercial Nylons (and different to our amorphous polyamide) as evidenced by its semi-crystalline nature (Supplementary Fig. 27). Further thermomechanical analysis of polymer films by dynamic mechanical analysis (DMA) confirmed the effect of the *cis-trans* ratio on $T_g$. (Fig. 2f; Supplementary Fig. 28).

Additional investigation of the mechanical properties for polymers with different stereochemistry was then conducted. Overall, the polyamides possessed high moduli with moderate

ductility. Specifically, the mechanical properties are comparable with Nylon 6,6, which is a useful biomaterial for numerous applications. The Young's modulus ($E$) of the materials ranged from $1052 \pm 74$ MPa (35% *cis*) to $1278 \pm 42$ MPa (82% *cis*), and stiffness increased accordingly with *cis* content; this trend was also very similar for glass transition temperature (Fig. 2b; Supplementary Fig. 31). It was found that there is a statistical difference ($P < 0.05$) between the elastic moduli of materials with very high (≥73%) *cis* content and those with low (≤46%) *cis* content. However, the elastic moduli differences between the 35 and 46% *cis* materials was found to be statistically insignificant. An inverse relationship is generally true for plasticity where the lowest strain at break ($\varepsilon_{break}$) was observed for P1 ($\varepsilon_{break} = 109 \pm 28$%). The strain at break for P4 ($\varepsilon_{break} = 138 \pm 87$%) was not drastically different from P1 and was found to be statistically insignificant. However, it should be noted that the P4 trials were less reproducible with one sample exceeding 200% elongation. The lack of reproducibility is also apparent when examining the relatively high standard deviations for the $E$ and $\sigma_{stress}$ (Table 1, entry P4). However, another sample with similarly high *trans* content (37% *cis*) was much more reproducible and had a significantly higher strain at break albeit the molecular mass of the sample was considerably greater (Supplementary Fig. 31).

**Shape memory properties.** Many implants would benefit from minimally invasive attributes to reduce surgical trauma and therefore time to healing. Shape memory behaviour could potentially provide a minimally invasive route for implantation by reducing the material footprint during surgery and it may also increase the implant's resilience to certain deformations or damage. Although shape memory alloys have been developed for hard tissue applications, they are beset by long-term biocompatibility issues[66], and most shape memory polymers for

biomedical use are more similar to soft biological tissues (e.g., hydrogels)[67,68]. The introduction of a more tuneable and dynamic non-resorbable biomaterial possessing excellent biocompatibility and robust mechanical properties could be extremely useful to provide a fine degree of control over the material's behaviour.

We discovered that all formulations displayed shape memory behaviour (Supplementary Tables 1 and 2), a rare phenomenon in polyamides that has previously only been observed in thermoset materials[24,69]. An important note to make is that these thermoplastics display good initial shape recovery, on par with cross-linked materials containing permanent net-points which help promote their original shape structures. In our system, it is likely that these features are a result of robust chain entanglements (due to high-molecular-weight) and entropic free-energy contributions combined with a high degree of hydrogen bonding among the amide moieties. We attempted to more precisely identify the origin of the shape memory behaviour by examining the surface morphology below and above the $T_g$ of P1 using AFM; however, no significant micro- or nano-scale morphological changes were visible (Supplementary Fig. 39). We are still probing the molecular emergence of the shape memory.

Regardless of stereochemistry, full strain recovery and strain fixation were achieved (Fig. 3; Supplementary Table 1). Following the trends revealed by thermomechanical analysis, high *cis* polymers displayed reduced strains (in line with the mechanical behaviour of the system). However, the high *cis* polymers were also more likely to undergo shape recovery when approaching the $T_g$ (at the increase of the dampening ratio as determined by tan δ), while the low *cis* polymers required prolonged isothermal equilibrium at 120 °C (well above the $T_g$) to achieve full strain recovery (Fig. 3b). The slower recovery of the low *cis* samples may be a consequence of the better polymer chain packing that is afforded by relative abundance of hydrogen bonding around individual chains. On the other hand, high *cis* polymers may have a decreased relative concentration of hydrogen bonding because of a more open molecular structure, and therefore reduced interactions with surrounding carbonyl moieties.

We also investigated the durability by performing cyclic shape memory experiments, including examining differences that could result from stereochemistry as well as choice of thiol comonomer (Supplementary Figs. 29 and 30, Supplementary Table 3). Regardless of composition, shape memory performance decreased after cycling (especially after ten cycles). This diminished cyclic repeatability in the shape memory behaviour is not unexpected for thermoplastic shape memory materials and may also be attributable to fatigue in the macroscopic film. Importantly, potential medical applications that make use of shape memory behaviour to produce minimally invasive devices do not require a material to behave as an actuator. Rather, a single shape-recovery response is necessary for minimally invasive surgeries. The performance of the polyamides displayed here meet these criteria and could be suitable in such applications.

**Biocompatibility studies.** Hydrolytic stability and water uptake behaviour were assessed for a high *trans* sample (33% *cis*, $M_w = 109$ kDa) and a high *cis* sample (86% *cis*, $M_w = 123$ kDa). The polymers were processed into disks via heated compression moulding, and the accelerated degradation profiles were assessed in triplicate in PBS, 1 M KOH and 5 M KOH solutions at 37 °C (Supplementary Fig. 36). For both polymers, minimal degradation and slight swelling were observed, irrespective of the medium, which indicated that this polymer class has robust hydrolytic stability. However, the swelling was more significant in the high *cis* samples (135 ± 5.7%, 1 M KOH) compared with the high *trans* samples (107 ± 4.4%, 1 M KOH). This trend is somewhat surprising since both materials are amorphous and have similar thermal profiles, but it is consistent with the hypothesis that stereochemically dependent shape memory recovery behaviour results from the closeness of packing of the polymer chains in the bulk material. Regardless of the medium, the high *trans* materials

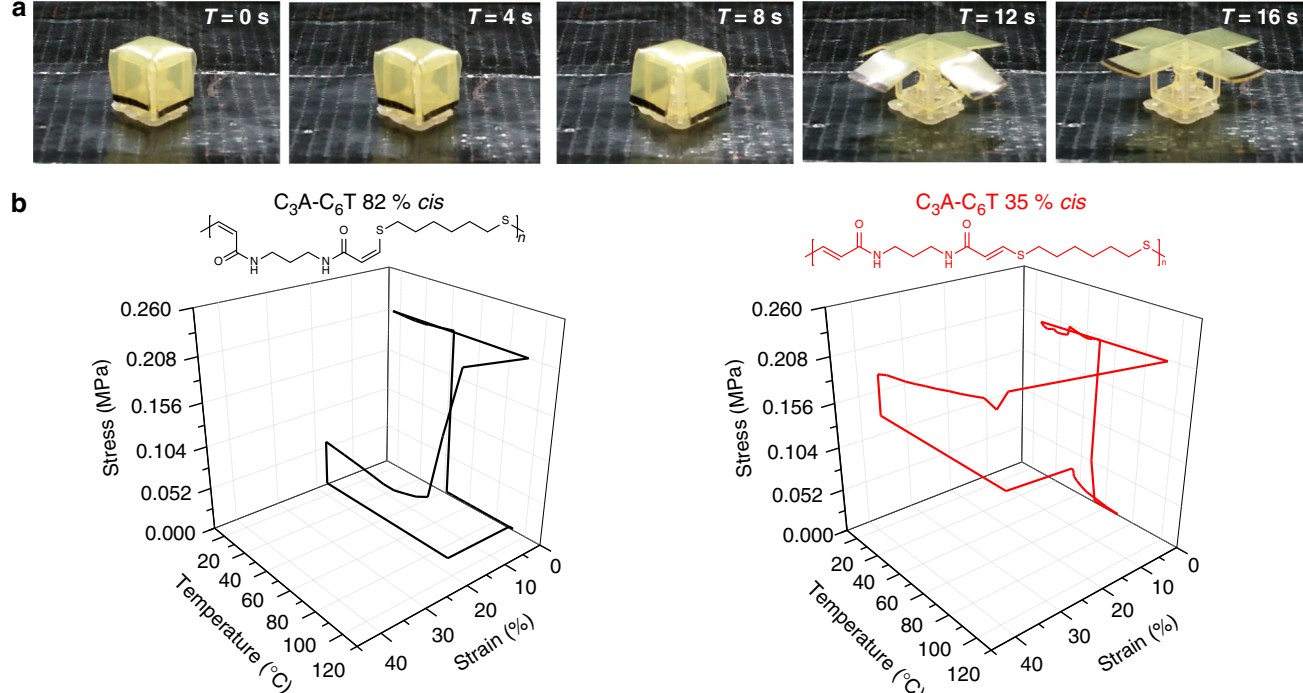

**Fig. 3 Shape memory behaviour. a** Visual demonstration of shape memory for $C_3A$-$C_6T$ 82% by heating to 120 °C. Note: Sample edges were darkened with ink to aid in visualising movement. **b** DMA analysis of shape memory behaviour for $C_3A$-$C_6T$ (high *cis* and low *cis*).

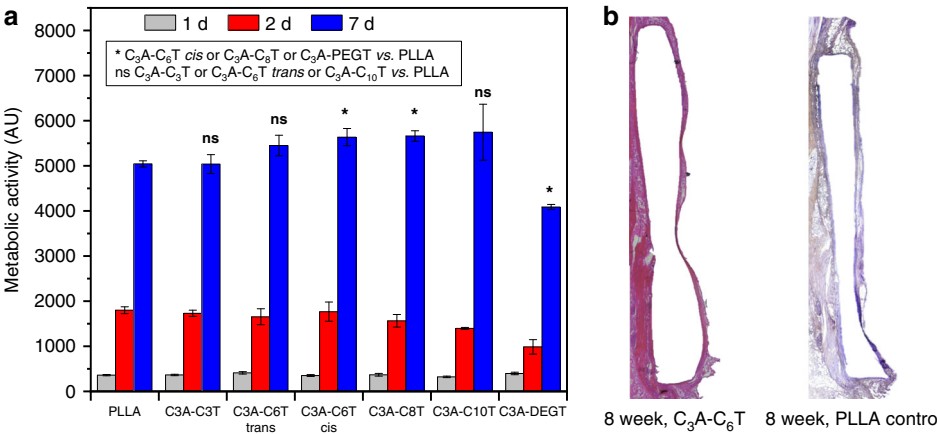

**Fig. 4 Cell biology and in vivo study. a** Cytocompatibility assays for all polyamides. Statistical analysis was performed using an ordinary two-way ANOVA test between the 7-day cell proliferation data on PLLA and each of the polymers synthesised ($P < 0.05$). Error bars represent 1 s.d. **b** Histological analysis of $C_3A$-$C_6T$ and PLLA implanted samples. PLLA was used as the control.

initially swelled to a maximum water uptake after 10 days followed by a small decrease in mass (relative to maximum), but the high *cis* materials continued to swell throughout the duration of the study. These results indicate that stereochemistry also affects water uptake behaviour, and subsequently the overall hydrolytic stability, which could also be leveraged in the future design of biomaterial devices and implants.

In order to investigate the potential for use in biomaterial applications, a proliferation assay was used as an initial method to determine cellular responses to all polyamides ($C_3A$-$C_6T$ *cis/trans*, or P1 and P4, and high *cis* polymers of other composition). Mouse osteoblasts (MC3T3-E1) were cultured in cell culture medium containing 10% foetal bovine serum on thin discs of the polyamide or on the control polymer, poly($L$-lactide) (PLLA). Cell proliferation was measured with a PrestoBlue metabolic assay, after 1, 3 and 7 days of incubation. This cell line is generally considered as a suitable cell model for studying material cytocompatibility[70–72]. After 7 days, the number of cells on each sample had increased dramatically, indicating good cytocompatibility (Fig. 4a). Statistical analysis ($P < 0.05$) revealed a superior cytocompatibility of $C_3A$-$C_6T$ *cis* and $C_3A$-$C_8T$ compared with the PLLA control, while $C_3A$-DEGT was found to support cell proliferation less than PLLA.

The in vivo biocompatibility of high *cis* $C_3A$-$C_6T$ (P1) was subsequently examined over 8 weeks in murine subcutaneous implantation studies. Hematoxyclin and eosin staining (H&E staining) was used to detect fibrous capsule granuloma formation and inflammatory cell presence, based upon ISO 10993-6 standards as assessor metrics, where individual slides were assessed for cell presence and numbers and assigned an inflammatory score at the tissue-material interfacial region (Table 2). Comparisons to PLLA controls indicated that the selected polyamide was both biocompatible as well as biostable. PLLA is considered a gold standard for implantable synthetic polymers and does not degrade over the time frame of the in vivo studies (8 weeks), hence we selected it as a suitable control for this experiment. Moreover, the biocompatibility of nylons has long been established in a variety of different animal models[73,74]. Capsule formation in both species was found to be <200 μm and uncalcified, which indicated that no severe inflammatory response occurred as a consequence of the presence of the material and is within the range of an accepted response for an implantable material (Fig. 4b). The absence of necrosis was a further indication of the biocompatibility of this polymer system, and the reduced macrophage presence outside the tissue capsule indicated low particulate formation and inflammation related to material

degradation. The presence of macrophages at the implant-tissue interface was expected and provided evidence of implant site healing. Similar macrophage loads at both the control and polyamide sites supported our claims of being a weakly immune-activating biomaterial, as PLLA is a well-known biomaterial used in a wide variety of implantable materials/devices across multiple species. Both materials displayed inflammatory cell levels and species that were consistent with low level inflammation, indicating good biocompatibility of the materials.

We also investigated the in vivo stability of the materials by comparing post-implanted samples (at 4-week and 8-week time points) against samples that had not been implanted. IR spectroscopic analysis of the samples indicated that the material composition was remarkably consistent before/after implantation as evidenced by the absence of any changes to their absorption spectra (Supplementary Fig. 37). However, we noted changes in the $^1H$ NMR spectra among the samples, particularly for the 8-week sample (Supplementary Fig. 38). We observed new low-field proton resonances ($\delta = 1.0 - 4.5$), but these may be explained by histological processing techniques (i.e., incomplete rinsing or cleaning of the sample). There were also minor variations in the vinylic resonance region ($\delta = 7.0 - 7.5$), suggesting some interaction or reactivity of the polymer in the biological media. Nevertheless, the IR spectra and physical appearance remained nearly identical among the samples. Finally, we observed reasonable maintenance of mechanical properties when samples were immersed in aqueous media as evidenced by immersion DMA experiments and tensile testing of "wet" samples (Supplementary Figs. 34 and 35). Together, these data suggest the materials are quite stable in application settings.

**Modulation of material properties by varying thiol comonomer.** After examining the $C_3A$-$C_6T$ polymers, other thiol comonomers were polymerised with $C_3A$ to examine the effect that the thiol comonomer had on thermomechanical properties of the resultant polymers (Table 3; Supplementary Figs. 11–20, Supplementary Notes 1–4). In order to control for the effects of stereochemistry, all polymers were targeted for high *cis* content (*ca.* 80%) using the conditions for the synthesis of P1 (Table 1, entry P1). Differential scanning calorimetry (DSC) of the high *cis* polymers also revealed the absence of a $T_m$, similar to P1-4 (Supplementary Fig. 24). Nevertheless, the $T_g$ displayed a positive correlation with % *cis* content. Unsurprisingly, as the number of carbon atoms increased in the thiol moiety ($C_3$, $C_6$, $C_8$ and $C_{10}$)

**Table 2 Analysis of fibrous capsule granuloma formation and inflammatory cell presence for $C_3A$-$C_6T$ and PLLA implanted samples using ISO 10993 standard scoring ($n = 6$, ± represents 1 s.d.).**

| Species | PLLA, 4 weeks | $C_3A$-$C_6T$, 4 weeks | PLLA, 8 weeks | $C_3A$-$C_6T$, 8 weeks |
|---|---|---|---|---|
| Neutrophils | 0.3 ± 0.6 | 0.2 ± 0.5 | 0.5 ± 0.7 | 0.6 ± 0.6 |
| Lymphocytes | 2.1 ± 0.5 | 2.3 ± 0.7 | 2.1 ± 0.6 | 2.0 ± 0.8 |
| Plasma cells | 0.1 ± 0.1 | 0.1 ± 0.1 | 0.1 ± 0.1 | 0.1 ± 0.2 |
| Mononuclear macrophage | 1.5 ± 0.7 | 1.3 ± 0.9 | 1.4 ± 0.9 | 1.1 ± 0.8 |
| Multinucleated foreign body giant cells | 0.0 ± 0.0 | 0.0 ± 0.0 | 0.0 ± 0.0 | 0.0 ± 0.0 |
| Necrosis | 0.0 ± 0.0 | 0.0 ± 0.0 | 0.0 ± 0.0 | 0.0 ± 0.0 |

**Table 3 Characterisation of $C_3A$-$C_XT$ polymers obtained from the polymerisation of $C_3A$ and various thiols using 1 mol% base at $[C_3A] = 1$ M.**

| Entry | Solvent | Base | %cis[a] | $M_w$ (kg mol$^{-1}$)[b] | $E$ (MPa) | $\varepsilon_{break}$ (%) | UTS (MPa) | $T_g$ (°C)[c] |
|---|---|---|---|---|---|---|---|---|
| $C_3A$-$C_3T_{(cis)}$ | DMSO/MeOH (2/1) | DBU | 82 | 92.7 | 1547.8 ± 44.4 | 24 ± 3 | 78.6 ± 1.1 | 107 |
| $C_3A$-$C_8T_{(cis)}$ | DMSO/MeOH (2/1) | DBU | 79 | 102.9 | 939.2 ± 67.0 | 217 ± 11 | 63.7 ± 3.7 | 86 |
| $C_3A$-$C_{10}T_{(cis)}$ | DMSO/MeOH (2/1) | DBU | 78 | 80.5 | 927.7 ± 26.7 | 313 ± 19 | 69.4 ± 3.7 | 71 |
| $C_3A$-$C_{DEG}T_{(cis)}$ | DMSO/MeOH (2/1) | DBU | 78 | 140.4 | 1623.7 ± 77.0 | 318 ± 70 | 78.1 ± 2.4 | 64 |

[a]%cis content determined by $^1$H NMR spectroscopic analysis.
[b]$M_w$ determined by SEC (DMF, 0.5% w/w NH$_4$BF$_4$) analysis against poly(methyl methacrylate) (PMMA) standards.
[c]Reported from the DSC thermograms of 2nd heating cycle. Mechanical data ($E$, $\varepsilon$-break and UTS) calculated from $n \geq 3$ samples. ± represents 1 s.d.

the $T_g$ decreased (Fig. 5b; Supplementary Fig. 25). The mechanical properties (such as $E$ and $\sigma_{stress}$) followed a similar trend and these differences among samples were found to be statistically significant (P < 0.05) (Fig. 5a; Supplementary Fig. 32). However, as the linker length increased the ductility of the materials also greatly improved ($C_3A$-$C_3T$, $\varepsilon_{break} = 24 \pm 3$% versus $C_3A$-$C_{10}T$, $\varepsilon_{break} = 313 \pm 19$%) and these differences were statistically significant as well. Importantly, no significant differences in shape-memory behaviour, either the strain fixation or the recovery stress, were observed among the compositions (Supplementary Table 2).

The most interesting mechanical properties were displayed by the $C_3A$-$C_{DEG}T$ polymer, which incorporated ethers in the repeat unit. The introduction of the diethylene glycol moiety into the polymer backbone resulted in a material with the lowest $T_g$ and we predicted this would lead to softer, yet more ductile material (Fig. 5b). In fact, $C_3A$-$C_{DEG}T$ was more ductile ($\varepsilon_{break} = 318 \pm 70$%) than other materials and was found to be statistically different (P < 0.05) from all compositions except for $C_3A$-$C_{10}T$. But remarkably, it also possessed the greatest modulus ($E = 1623 \pm 77$ MPa) and yield stress ($78.1 \pm 2.4$ MPa) for the entire polyamide series (Supplementary Table 4), although these values are not statistically different from the $C_3T$ or $C_6T$ polymers. A similar enhancement of mechanical properties has previously been observed in poly(ether-thioureas)[75] as compared to poly(alkyl-thioureas). The authors proposed that the extra H-bond acceptors (ethereal oxygen atoms) increased the H-bonding interactions between the polymer chains without inducing crystallisation. Our results further suggest that incorporating ethereal oxygen atoms into polymers displaying significant hydrogen bonding (e.g., polyamides or polythioureas) may present a path to processable materials with high toughness. Furthermore, by decreasing the $T_g$, the transition temperature of the shape memory was closer to a biologically compatible temperature (Supplementary Fig. 28 and Supplementary Table 2).

## Discussion

In conclusion, the nucleophilic thiol-yne addition was exploited to synthesise a family of stereochemically controlled polyamides.

The thermomechanical properties were tuned either by modulating the stereochemistry of the alkene moiety in the backbone or changing the thiol comonomer. Materials with high *cis* content were stiffer and less ductile than high *trans* polymers, but this was not correlated with crystallinity since all polymers appeared to be amorphous. In vitro cytocompatibility and in vivo assessments indicated that these materials possess excellent stability and biocompatibility, which suggests that they could serve as adaptive, non-resorbable biomaterials. Nylons have traditionally been some of the most versatile biomaterials due to material property tuning which may be achieved solely through compositional adjustment; however, the system presented offers multiple possibilities for precise property modulation. Our polyamides have comparable mechanical performance to nylons but are strikingly more adaptive and easier to process. The unique combination of biostability and modifiable thermomechanical properties, in addition to the aging behaviours revealed during shape-memory testing, open avenues to durable biomedical devices such as arterial grafts, heart valves and sewing cuffs. In these applications, the tuneable mechanical properties of the materials presented here, along with their stability, would allow for consistent, life-long performance after implantation.

## Methods

**General.** All manipulations of air-sensitive compounds were carried out under a dry nitrogen atmosphere using standard Schlenk techniques. All compounds, unless otherwise indicated were purchased from commercial sources and used as received. The following chemicals were vacuum distilled prior to use and stored in Young's tapped ampoules under N$_2$: 1,3-propanedithiol ($C_3T$) (Sigma-Aldrich, 99%), 1,6-hexanedithiol ($C_6T$) (Sigma-Aldrich, ≥97%), 1,8-octanedithiol ($C_8T$) (Alfa Aesar, 98%), 1,10-decanedithiol ($C_{10}T$) (Alfa Aesar, 95%), 2,2-(ethylene-dioxy)diethanethiol ($C_{DEG}T$: Sigma-Aldrich, 95%). All solvents for recrystallisation were used as received. All measurements were taken from distinct samples, unless otherwise noted.

**Synthesis of dipropiolamide monomer ($C_3A$).** A 3-neck 500-mL round bottom flask fitted with a 100-mL dropping funnel was charged with 1,3-diaminopropane (19.5 mL, 232 mmol, 1.00 equiv) and 100 mL water. The solution was cooled to 0 °C using an ice bath and then methyl propiolate (40.0 g, 476 mmol, 2.05 equiv) was added over 30 min using a dropping funnel. During the course of the addition, the mixture became slightly yellow and a precipitate formed. After the addition, the resultant mixture was stirred for a further 6 h at 0 °C. Note: we observed undesirable by-products and lower reaction yields when the reaction temperature was

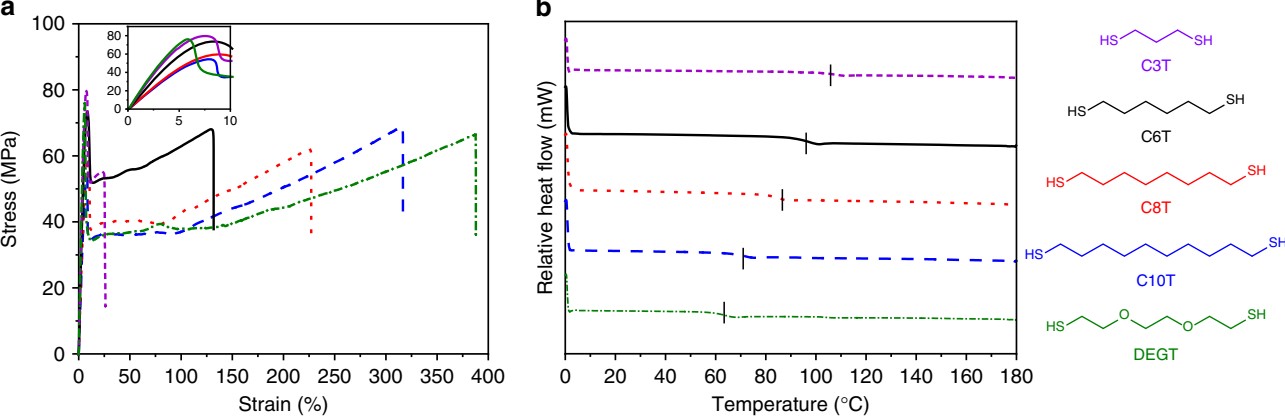

**Fig. 5 Effect of thiol comonomer on thermomechanical behaviour. a** Exemplary stress vs strain curves tested at 10 mm min⁻¹, 22 °C. **b** DSC thermograms of 2nd heating cycle tested at 10 °C min⁻¹. Position of $T_g$ for each polymer sample is indicated with a vertical hash mark.

ca. >5–10 °C. The precipitate was collected using vacuum filtration and washed with cold water ($5 \times 25$ mL) to afford a light-yellow solid. After drying on the benchtop at 22 °C for 24 h to remove water, the compound was purified by recrystallisation (cooling to −20 °C for ca. 16 h after heating) from a mixture of chloroform/methanol (5/1) to afford the title compound as a pale-yellow solid (13.9 g, 34%). ¹H NMR (400 MHz, DMSO–$d_6$) δ 8.69 (t, $J = 5.8$ Hz, 2H), *8.17 (t, $J = 5.4$ Hz), *4.43 (s), 4.11 (s, 2H), *3.24 (q, $J = 6.8$ Hz), 3.06 (q, $J = 6.6$ Hz, 4H), 1.55 (p, $J = 7.0$ Hz, 2H). ¹³C NMR (101 MHz, DMSO–$d_6$) δ 151.58, *81.27, 78.31, 75.54, 36.66, *36.36, *29.73, 28.27. * = minor signals due to rotamers. HRMS (ESI-TOF) (m/z): $[M + H]^+$ calculated for $C_9H_{10}N_2NaO_2$, 201.0633; found, 201.0634.

**Representative polymerisation of C₃A-C₆T polymers**. A 100-mL round bottom flask was charged with C₃A (4.00 g, 22.4 mmol, 1.018 equiv) and a separate 20-mL scintillation vial was charged with C₆T (3.31 g, 22.1 mmol, 1.000 equiv). The thiol was quantitatively transferred to the round bottom flask using DMSO (45 mL) for a final monomer concentration of ~0.5 M. The reaction mixture was placed in a water bath (ca. 15 °C) and 1,8-diazabicyclo(5.4.0)undec-7-ene (DBU) (32.9 μL, 0.22 mmol, 0.01 equiv) was injected in one portion. The addition of DBU produced an exotherm which was mitigated by heat transfer to the water bath. After 2 min of stirring, the reaction flask was sealed, removed from the bath and stirred at 50 °C in order to increase the solubility of the polymer product. After 2 h the reaction was quenched with 1-dodecanethiol (0.5 mL, 2.1 mmol) to end-cap any alkyne chain-ends and stirred for 30 min. Then, the solution was diluted with DMSO (ca. 50 mL) and BHT (0.5 g, 2.2 mmol) was added in order to prevent cross-linking during the precipitation step. The reaction mixture was then precipitated into methanol (1000 mL), and the polymer was collected by decanting the supernatant. The polymer was stirred in methanol (200 mL) for ca. 12 h to help remove residual DMSO before drying in vacuo (500 mTorr) at 120 °C overnight (ca. 16 h). GPC analysis (DMF + 5 mM NH₄BF₄) $M_p$ = 96.8 kDa, $M_w$ = 131.4 kDa, $M_n$ = 38.8 kDa, kDa, $Đ_m$ = 3.39. ¹H NMR (400 MHz, DMSO–$d_6$) % cis = 73 δ 7.81 (t, $J = 5.7$ Hz, cis + trans overlapping, 2H), 7.32 (d, $J = 15.0$ Hz, trans), 6.89 (d, $J = 10.1$ Hz, cis 2H), 5.89 (d, $J = 15.0$ Hz, trans), 5.84 (d, $J = 10.0$ Hz, cis 2H), 3.14–3.02 (m, cis + trans overlapping, 4H), 2.79 (t, $J = 7.3$ Hz, trans), 2.63 (t, $J = 7.3$ Hz, cis 4H), 1.64–1.47 (m, 6H), 1.43–1.28 (m, 4H). ¹³C NMR (101 MHz, DMSO–$d_6$) δ 165.60$_{cis}$, 163.71$_{trans}$, 143.72$_{cis}$, 139.25$_{trans}$, 117.83$_{trans}$, 115.81$_{cis}$, 36.46$_{tram}$, 36.23$_{cis}$, 34.70, 31.05$_{trans}$, 30.00$_{cis}$, 29.42$_{cis}$, 28.38$_{trans}$, 27.64$_{trans}$, 27.42$_{cis}$.

**General polymerisation of high cis content polymers**. A 100-mL round bottom flask was charged with C₃A (4.00 g, 22.4 mmol, 1.018 equiv), and a separate 20 mL scintillation vial was charged with the corresponding dithiol (22.1 mmol, 1.000 equiv). The dithiol was quantitatively transferred to the round bottom flask containing C₃A by washing with DMSO (30 mL) and methanol (15 mL) for a final monomer concentration of ~0.5 M. The reaction mixture was placed in a water bath (ca. 15 °C), and DBU (32.9 μL, 0.22 mmol, 0.01 equiv) was injected in one portion. The addition of DBU produced an exotherm which was mitigated by heat transfer to the water bath. After 2 min of stirring, the reaction flask was sealed, removed from the water bath and stirred at 50 °C in order to increase the solubility of the polymer product. After 2 h, the reaction was quenched with 1-dodecanethiol (0.5 mL, 2.1 mmol) to end-cap any alkyne chain-ends and stirred for 30 min. Then, the solution was diluted with DMSO (ca. 50 mL), and BHT (0.5 g, 2.2 mmol) was added in order to prevent cross-linking during the precipitation step. The reaction mixture was then precipitated into methanol (1000 mL), and the polymer was collected by decanting the supernatant. The polymer was stirred in methanol (200 mL) for ca. 12 h to help remove residual DMSO before drying in vacuo (500 mTorr) at 120 °C overnight (ca. 16 h).

**NMR spectroscopy**. NMR spectroscopy experiments were performed at 25 °C on a Bruker DPX-400 NMR instrument equipped operating at 400 MHz for ¹H (100.57 MHz for ¹³C). ¹H NMR spectra are referenced to residual proton solvent (δ = 2.50 for DMSO–$d_5$), and ¹³C NMR spectra are referenced to the solvent signal (δ = 39.52 for DMSO–$d_6$). The resonance multiplicities are described as s (singlet), d (doublet), t (triplet), q (quartet) or m (multiplet).

**Mass spectrometry**. High-resolution electrospray ionisation mass spectrometry was performed on a Bruker MaXis Plus using a TOF detector in the Department of Chemistry, University of Warwick, CV4 7AL, UK.

**Size-exclusion chromatography (SEC)**. SEC measurements were performed on an Agilent 1260 Infinity II Multi-Detector GPC/SEC System fitted with RI and ultraviolet (UV) detectors ($λ = 309$ nm) and PLGel 3 μm ($50 \times 7.5$ mm) guard column and two PLGel 5 μm ($300 \times 7.5$ mm) mixed-C columns with DMF containing 5 mM NH₄BF₄ as the eluent (flow rate 1 mL/min, 50 °C). A 12-point calibration curve ($M_p$ = 550–2,210,000 g mol⁻¹) based on poly(methyl methacrylate) standards (PMMA, Easivial PM, Agilent) was applied for determination of molecular weights.

**Thermogravimetric analysis (TGA)**. TGA thermograms were obtained using a TGA/DSC 1-Thermogravimetric Analyzer (Mettler Toledo). Thermograms were recorded under an N₂ atmosphere at a heating rate of 10 °C min⁻¹, from 50 to 500 °C, with an average sample weight of ca. 10 mg. Aluminium pans were used for all samples. Decomposition temperatures were reported as the 5% weight loss temperature ($T_{d\,5\%}$).

**Mechanical property measurements (film preparation and uniaxial tensile testing)**. Thin films of each polymer were fabricated using a vacuum compression machine (TMP Technical Machine Products Corp.). The machine was preheated to 170 °C. Then polymer was added into the $50 \times 50 \times 1.00$-mm mould, and put into the compression machine with vacuum on. After 15 min of melting, the system was degassed three times. Next, 10 lbs*1000, 15 lbs*1000, 20 lbs*1000, 25 lbs*1000 of pressure were applied consecutively for 2 min. each. After that, the mould was cooled down with 1000 psi of pressure to prevent wrinkling of the film's surface. The films were visually inspected to ensure that no bubbles were present in the films. Dumbbell-shaped samples were cut using a custom ASTM Die D-638 Type V. The dimensions of the neck of the specimens were 7.11 mm in length, 1.70 mm in width and 1.00 mm in thickness.

Tensile Tests were conducted at 10 mm min⁻¹ using dumbbell-shaped samples that were prepared using the method stated above. Tensile tests were carried out using an Instron 5543 Universal Testing Machine or Testometric M350-5 CT Machine at room temperature (25 ± 1 °C), unless otherwise noted. The gauge length was set as 7 mm, and the crosshead speed was set as 10 mm min⁻¹. The dimensions of the neck of the specimens were 7.11 mm in length, 1.70 mm in width and 1.00 mm in thickness. The elastic moduli were calculated using the slope of linear fitting of the data from strain of 0–0.1%. The reported results are average values from at least three individual measurements unless otherwise specified.

**Differential scanning calorimetry (DSC)**. The thermal characteristics of the polymer thin films were determined using differential scanning calorimetry (STARe system DSC3, Mettler Toledo) from 0 to 180 °C at a heating rate of 10 °C min⁻¹ for three heating/cooling cycles unless otherwise specified. The glass transition temperature ($T_g$) was determined from the inflection point in the second heating cycle.

**Dynamic mechanical analysis (DMA).** Dynamic mechanical thermal analysis (DMTA) data were obtained using a Mettler Toledo DMA 1-star system and analysed using the software package STARe V13.00a (build 6917). Thermal sweeps were conducted using bar-shaped films (11.00 mm in length × 6.00 mm in width x 1.00 mm in thickness) cooled to −80 °C and held isothermally for *ca.* 5 minutes. Storage and loss moduli, as well as the loss factor (ratio of $E''$ and $E'$, tan δ), were probed as the temperature was swept from −80 to 150 °C, 2 °C min$^{-1}$. Thermo-mechanical behaviour was determined for $n = 3$ samples in this way. Shape-memory behaviour of the same films was examined by first heating as-processed materials to 120 °C, holding samples isothermally for 8 min, and deforming samples using a 1 N load that was held static as the sample was cooled again to ambient conditions (*ca.* 25 °C). Strain fixation was calculated as the strain held by the material as the load was released at ambient temperature. Deformed samples were then heated at 10 °C min$^{-1}$ up to 120 °C, where the samples were held isothermally for 20 min to allow for total shape recovery to take place. The temperature at which strain recovery began was labelled as the recovery temperature, and the final recovered strain was the recoverable strain for the materials. Sample behaviour was averaged for $n = 3$ films.

**Cyclic shape memory.** Performance evaluation was conducted using bar-shaped films (0.5 mm × 6 mm × 20 mm). The sample was first heated to 20 °C above the $T_g$ of the material, where samples were held until equilibrated. At this point, the proximal film end was fixed, and its distal end was bent, with the film uniformly bent around a central shaft until the distal end was parallel to the proximal. The sample was mechanically restrained to fix the film in this temporary shape until it had cooled to 22 °C. At this point, the mechanical constraint was removed, and strain fixation was measured optically over a 5-min period. The sample was then heated to 20 °C above the $T_g$ of the material, where samples were held until equilibrated, with the shape recovery measured optically at 5 s intervals. Final recovered strain was determined as a function of recovered deformation of the distal end compared with the fixed proximal end. This was repeated for 60 cycles for each film, with $n = 3$ films per examined species.

**Atomic force microscopy (AFM).** AFM was performed on a JPK Nanowizard 4 system fitted with a heater-cooler module to control sample temperature (24–90 °C) during imaging. Images were acquired in the supplied acoustic enclosure and with vibration isolation, using a Nanosensor PPP-NCHAuD tip with a force constant of around 42 N m$^{-1}$. For data acquisition and handling Nanowizard Control and Data Processing Software V.6.1.117 in QI mode with a set point of 100 nN and pixel time of 8 ms was used. In this mode, a force/distance curve is collected for each pixel in the acquired image. Adhesion was calculated by first subtracting the baseline from each force curve and then identifying the lowest point of the retraction force curve. Slope, a proxy for the Young's modulus of the material, was taken as the average slope of the steepest segment of the approach force curve. On changing the temperature, the sample was allowed to equilibrate for 15 min before commencing imaging. To calibrate the tip for use at different temperatures (and therefore allow meaningful comparison of force data), following equilibration at the desired temperature the tip was positioned 50 μm above the surface and automatically calibrated using the built-in Sader method in the Nanowizard software, with the temperature set to that of the heating stage.

**Hydrolytic degradation studies.** Accelerated degradation studies were conducted under conditions previously reported by Lam et al[76]. All polymers ($n = 3$ for each composition in each solution, 18 total samples) were prepared as "degradation disks" (*ca.* 0.1 g) using heated compression moulding and subjected to aqueous environments of various pH (PBS, 1 M aq. NaOH, and 5 M aq. NaOH). The disks were placed in individual vials containing 20 mL of the corresponding solution and incubated at 37 °C with constant agitation at 60 rpm. Before analysis, the surface of each disk was dried in order to remove excess water before the weight was peri-odically measured using an analytical balance.

**In vitro cell studies.** Polyamides were moulded into thin discs (*ca.* 0.3-mm thickness) with flat surfaces using compression moulding at 170 °C between two glass slides. PLLA was spin-coated onto a glass slide from a chloroform solution. MC3T3-E1 cells (ATCC, CRL-2593) were seeded at a density of 4000 cells cm$^{-2}$ on top of the polymer discs and cultured at 37 °C and 5% CO$_2$ in α-MEM enriched with 10% FBS and 1% pen/strep. Viability was detected after 24 h, 72 h and 7 days of incubation using PrestoBlue proliferation assay and following manufacturer instructions. Fluorescence ($\lambda_{Ex.} = 530$ nm, $\lambda_{Em.} = 590$ nm) was measured using a BioTek plate reader. Statistical analysis was performed using an ordinary two-way ANOVA test.

**In vivo animal studies.** Experiments were performed in accordance with the European Commission Directive 2010/63/EU (European Convention for the Pro-tection of Vertebrate Animals used for Experimental and Other Scientific Pur-poses) and the United Kingdom Home Office (Scientific Procedures) Act (1986) with project approval from the institutional animal welfare and ethical review body (AWERB). Anaesthesia was induced in adult male Sprague Dawley rats ($n = 6$ for each time point, 200–300 g) with isoflurane (2–4%; Piramal Healthcare) in pure

oxygen (BOC). Animals were placed prone onto a thermo-coupled heating pad (TCAT 2−LV; Physitemp), and body temperature was maintained at 36.7 °C. Four incisions of 2-3 cm were made: two bilaterally over the spino-trapezius in equal location, and two bilaterally over the medial aspect of the dorsal external obliques. The skin was separated from the muscle with large forceps, and any excess fat was removed. The implants were tunnelled under the skin and placed in direct contact with the muscle, at sites distal to the incision. The order of the implants was randomised but constrained so that each implant appeared in each location at least once, including the PLLA control disc. All rats were implanted with a control material (PLLA). The wounds were sealed with a subcuticular figure of 8 purse string suture with a set-back buried knot using 3-0 polyglactin sutures (Ethicon—Vicryl Rapide™). The surgical procedure was performed under the strictest of aseptic conditions with the aid of a non-sterile assistant. Post-surgical analgesia was administered, and rats were placed into clean cages with food and water ad libitum. 6 animals were sacrificed at each of the selected time points (4 and 8 weeks).

Histology Studies Samples for histology were collected post-mortem by cutting the area around the polymer implant to obtain tissue samples of *ca.* 1.5 cm$^2$ with the polymer disc in the middle. Samples ($n = 6$) were then cut in the middle of the polymer disc to get a cross-section of both polymer and tissue, then placed in a 4% PFA solution overnight at 4 °C for fixation. Fixed samples were then washed with increasing % of ethanol (70–100%) followed by 100% xylene before embedding in paraffin. Embedded samples were cut to 5–20-μm-thick slices, de-paraffined in histological grade xylenes, rehydrated, and stained with either hematoxylin and eosin (H&E) or Masson's Trichrome stain, following standard Sigma-Aldrich protocols. Brightfield microscopy was used to examine pathology of select samples (at 4 weeks and 8 weeks), with inflammatory cell load, necrosis presence and capsule thickness analysed at discreet intervals around the sample perimeter. Inflammatory scoring was based on ISO 10993 standard scoring (scale range of 0–4, 0 = none, 1 = rare, 2 = mild, 3 = moderate, 4 = severe).

**Reporting summary.** Further information on research design is available in the Nature Research Reporting Summary linked to this article.

## Data availability
The authors declare that full experimental details and characterisation of materials are available in the Supplementary Information. All raw data that support the findings in this study are available from the corresponding authors upon request.

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

## Acknowledgements

J.C.W. and A.C.W. acknowledge funding from the European Union's Horizon 2020 research and innovation programme under the Marie Sklodowska-Curie grant agreement Nos. 751150 and 793247 respectively. M.L.B. acknowledges the W. Gerald Austen Endowed Chair in Polymer Science and Polymer Engineering for funding these efforts. A.P.D. thanks the European Research Council (grant number 681559) for funding. All three external reviewers are thanked for their time and contribution to the final version of this publication.

## Author contributions

J.C.W., A.P.D. and M.L.B. conceived the material synthesis and designed the project idea. J.C.W. synthesised and characterised the materials. J.C.W., J.Y. and A.C.W. performed thermal and mechanical analyses. T.R.W. conducted AFM analysis under the supervision of R.K.O'R. M.C.A. and A.P.D. designed the in vitro and, with R.T.R.H., in vivo experiments. M.C.A. performed in vitro analyses, M.C.A. and R.T.R.H. performed in vivo experiments while A.C.W. performed the histology. The paper was written through contributions of J.C.W., A.C.W., M.L.B. and A.P.D. All authors have given approval to the final version of the paper.

## Competing interests

The authors declare no competing interests.
