## [Peer Review File · Nature Communications]

Reviewers' comments:

Reviewer #1 (Remarks to the Author):

The authors describe an interesting study that claims on the development of 'Biocompatible and Elastomeric Polyamides with Stereochemically-Tunable Mechanical Properties'.

The usage of polyamides for (bio)medical applications is a very important aim and the manuscript is well written. The results will be of interest to others in the community and the wider field.

However, the publication in a more specialized journal might be considered.

In general, some revisions should be done prior to publication:

- The advantages of these polymers in comparison to other biomaterials should be emphasized more clearly.
- The manuscript should be placed into a larger context, and studies on further stereoregular polyamides (e.g. based on tartaric acid or on terpenes) should also be cited.
- SEC elugrams should also be shown in the main text (not only in SI).
- Additional figures of the cells should be shown.
- The possible reactivity of the alkene moiety in the biological environments after implantation should be discussed more detailed.

Reviewer #2 (Remarks to the Author):

The manuscript by Dove and Becker entitled "Biocompatible and Elastomeric Polyamides with Stereochemically-Tunable Mechanical Properties" describes a series of unsaturated polyamides that are extremely tough, possess highly tunable mechanical properties, and are biocompatible. The manuscript is well written and data support the conclusions. The observation of alkene stereochemistry dictating bulk properties such as tensile strength, modulus, and glass transition of the resulting elastomers is interesting and noteworthy.

I recommend publication after the following changes are made and comments addressed.

Major

A. In particular the paper has no statistical analysis of the data. Please provide statistics and state in the manuscript that the changes observed are significant or not.

B. Also a control 8 week in vivo experiment with nylon 6 or nylon 6,6 is required as the paper compares the performance of these new nylon polymers with nylon 6 and nylon6,6 in all other experiments except for the in vivo one.

C. Please perform a cyclic shape memory experiment and provide the results in the manuscript or SI.

All Comments

1. Page 1, line 12 – please remove the phrase long-term. Long-term biocompatibility is 1 or 2 years not 8 weeks. For example PCL.

2. Page 4, Table 1. Please add nylon 6 and nylon 6,6 to the table as this is the control polymers used to compare performance of the new polymers, as shown in Figure 2

3. Page 5. Please provide statistics for the data collected on the Young's modulus. Are these values different from one another? Similarly for the stress vs strain curves are the data collected at different temperatures different? It appears so – but further analysis is required.

4. Page 6: Line 125: from 0 – 280 °C, 10 K min⁻¹ 125 . Please be consistent in measurement units – K or °C.

5. Page 7: Line 168: A very long sentence – please rewrite as this is an important concept for the reader. Also please do not use “this” as a subject and noun to start a sentence.
6. Page 7: line 171: Please perform a cyclic shape memory experiment and provide the results in the manuscript or SI. This is an important performance characteristic of the material and needs to be known.
7. Page 8: Line 200: Please justify the choice of a particular cell line: Mouse osteoblasts (MC3T3-E1). Typical biocompatibility studies use NIH 3T3 cells.
8. Page 9: Figure 4a and Line 203: excellent cytocompatibility – what does the word excellent refer to in this statement? The results were similar to controls. Or is this not correct? Why is there increased metabolic activity with the polymer present or are the results similar to the control. Please provide some statistical analysis.
9. Page 9: line 212: PLLA is a good control for biocompatibility but it is not a nylon structure. Please repeat the experiment with nylon 6 or nylon 6,6 and provide the data in the manuscript.
10. Page 11: Line 253: Please check the sentence structure and fix grammatical error: may more processable materials
11. Page 13 Line 334: Please include page numbers for the ref 18. Surgery 22, (2011) ????

Supplemental Information document:

1. Line 5: ±- superscript
2. Lines 4 and 5: sometimes the sign is before comma and sometimes it's after. Please fix. The following sequence “, §”
3. What's the gender of mice (n = 6) studied?
4. Starting from line 235, adjust titles of spectra and maybe fit two spectra for each compound and their titles into one page.
5. Line 455: is there any protocol for animal work that you may have followed that you should reference?

Reviewer #3 (Remarks to the Author):

The manuscript by Worch et al. entitled “Biocompatible and Elastomeric Polyamides with Stereochemically-Tunable Mechanical Properties” presents the synthesis of a new polyamide series that offers interesting mechanical, shape memory, and in vivo compatibility properties for potential use as a biomaterial. The manuscript is generally well written and easy to read. This reviewer suggests major revisions for this manuscript.

Major points:

- 1) How and why were monomers chosen for this study? What is the rationale for the number of methylenes (linker length, etc.) for these initial structures that were studied? While certainly the stereochemistry plays a role, is there a point where that does not matter (when the monomer

lengths are sufficiently) increased? The authors do examine in lesser detail derivatives of the thiol monomers (of different numbers of methylenes and an ethylene oxide derivative) but did not examine derivatives of the alkyne – which is important to understand how further alterations in the chemical properties may enhance, dilute/diminish the stereochemical effects. This also can alter Tg and crystallinity as the methylene number in nylons on both sides of the amides are very important to physical properties (particularly noncovalent interactions between the polymer structures).

2) At the very least, it is important to include a synthetic analog to nylon 6,6 (Nylon 6 and 6,6 are used in Figure 2 but nylon 6, 6 would be needed if a 6, 6 analog is made). It would also be important to have a better a nylon control (3,14) that is more analogous to the lengths between the amides of the synthetic analog that is studied in details. Also, the authors attribute the shape memory properties to the H-bonding and again, it is important to have more accurate analogous nylons to compare the physical/shape memory properties.

3) It is not understood why all of the chemical analogs at the end of the manuscript (and their subsequent physical properties) were not included in the original Table 1. It would be more useful to have all of the analogs displayed side-by-side to further gain insight into the role of linker length. Why was the C3A-C6T analog selected for the extensive studies and not the ethylene oxide derivative? This was not explained in the manuscript.

4) What is the mechanism of shape memory? Discerning this is central to the impact of this manuscript. How/why does the ethylene oxide derivative exhibit shape memory?

5) How is the bulk structure of the materials impacted by stereochemistry? Scattering experiments could be completed to determine this as this could give insight into the mechanism of shape memory.

6) One cycle of shape memory is shown for each derivative. How many cycles can these materials handle before fatigue?

Minor points:

7) The chemical structures in Figure 1 are a bit small, the images should be increased in size.

8) It would be useful to add the Tg data to Table 1

9) It is important to have more discussion about the importance of the modulus (and comparison to other known materials) such that the general readership of Nature Communications can understand the relevance of these data compared to common materials.

10) PLLA was used as a control for the in vivo studies. Another control should likely be analogous nylons. Consistent controls should be used for the studies throughout this work.

11) There are a few typos in the paper that should be fixed.

Reviewers' comments:

Black text = reviewer comment

Blue text = author response

Reviewer #1 (Remarks to the Author):

The authors describe an interesting study that claims on the development of 'Biocompatible and Elastomeric Polyamides with Stereochemically-Tunable Mechanical Properties'.

The usage of polyamides for (bio)medical applications is a very important aim and the manuscript is well written. The results will be of interest to others in the community and the wider field. However, the publication in a more specialized journal might be considered.

In general, some revisions should be done prior to publication:

- The advantages of these polymers in comparison to other biomaterials should be emphasized more clearly.

Polyamides have been a biomaterial of choice for decades, and have found use as sutures, materials for ligament and tendon repair, catheter balloons, and membranes. The diversity of this list speaks to the versatility of polyamides, where their unique combination of low degradability (high stability), robust mechanical properties, and semi-crystallinity is used to maximize their performance both inside and out of the body. While many other material systems can find use in any one of these fields, for example polylactic acid in sutures, most materials are not suitable for such an application range. Like polyurethanes, polyamides may also be imbued with segmentation to further enhance mechanical performance. With the presented polyamide system, this behaviour is tunable through either stereochemistry or through compositional control, giving an additional measure of tunability for these biomaterials. We have added text into the introduction to address this concern and highlight the advantages of these materials. We believe that the second p/g of the introduction mostly addresses this and now reads:

"Synthetic polymers have been used in medical devices for more than five decades and recent key advances have largely focused on the development of biodegradable (or resorbable) materials for tissue applications.⁷ As such, the continued innovation of long-lasting non-resorbable polymers in joint and/or bone therapies, for example, are lagging behind and suffer from notable limitations such as wear, difficult processing, sterilization or high cost. Polyamides have been a biomaterial of choice for decades and they have been extensively developed for applications ranging from use as sutures^{9,10} or membranes¹¹ to vascular applications¹² because of their toughness, low-cost and outstanding biocompatibility.⁸ Polyamides are also widely used in bone engineering as a consequence of the materials' high strength and flexibility which is due to the extensive degrees of both crystallinity and hydrogen bonding.¹³⁻²¹ Even though this may be useful in selected orthopedic or vascular applications, these types of materials are typically difficult to process and functionalize, which has ultimately limited their performance in other applications. An ideal durable biomaterial platform would incorporate the thermomechanical and biological performance of polyamides while displaying enhanced processability and advanced functionality, such as shape memory for minimally invasive device designs."

Also, the last p/g of the introduction has the following sentences that addresses this concern:

"...The presence of both amide and unsaturated moieties along the backbone imbued the materials with unique properties, namely shape memory behavior and stereochemically-dependent thermomechanical properties...These properties make the polymers unique among polyamides, such as Nylons, which are semi-crystalline (when stretched) and display no shape memory behavior, factors which limits their utility as the medical device community begins to embrace minimally invasive designs..."

- The manuscript should be placed into a larger context, and studies on further stereoregular polyamides (e.g. based on tartaric acid or on terpenes) should also be cited.

We would like to thank the reviewer for bringing this to our attention, since it is certainly relevant to the manuscript. We have added the following sentences and references in the introduction: “Synthetic polyamides possessing chiral centers are comparatively abundant and well represented in the literature with the stereochemistry usually derived from the use of naturally-occurring co-monomers and/or precursors such as sugars⁴⁰⁻⁴⁸, amino acids^{49,50}, tartaric acid⁵¹⁻⁵³ and terpenes.⁵⁴⁻⁵⁶. The defining feature among the stereo-defined polyamides is their inherent crystallinity which has been characterized and assessed in most studies. However, polymer mechanical properties and how those relate to stereochemistry are generally absent from these studies, presumably due to the challenges synthesising high molecular weight step-growth polymers without harsh reaction conditions.”

- SEC elugrams should also be shown in the main text (not only in SI).

We have now included the SEC chromatograms of polymers possessing varied stereochemistry in Figure 1.

- Additional figures of the cells should be shown.

In the *in vitro* experiments, cells were seeded on top of 0.3 mm thick polymer films and therefore it was not possible to image cells directly on top of the polymer scaffolds because they are too thick. Normally, imaging can be achieved by spin coating the polymer on glass slides and subsequently seeding cells on top of the polymer layer; however, on account of the poor solubility in most solvents it was not possible to obtain a uniform layer of adequate thickness with these polymers even when spin coating from DMSO.

- The possible reactivity of the alkene moiety in the biological environments after implantation should be discussed more detailed.

(GPC, NMR, IR of implanted samples)

We thank the reviewer for raising this excellent point. To address it, we performed NMR and IR spectroscopic analysis (the solubility in DMF was poor for these implanted samples so we could not perform GPC analysis) of the implanted samples – 1 & 2 month time points. These data are described in the text (end of “Biocompatibility Studies” section) and figures found in the Supporting Information (Figure S37-38).

IR spectra of the implanted films were very similar to the pre-implanted sample (Figure S34). Importantly, there was little to no change in the alkene region ($1600-1700\text{ cm}^{-1}$) among all samples. However, the ^1H NMR analysis of the implants did reveal some slight changes to the polyamide structure, especially for the 2 month sample (Figure S35). In addition to new low-field resonances (of which some may be attributable to histological processing), we also observed some new vinylic resonances in the 2 month sample which suggests there is some interaction between the polyamide and biological media.

Reviewer #2 (Remarks to the Author):

The manuscript by Dove and Becker entitled “Biocompatible and Elastomeric Polyamides with Stereochemically-Tunable Mechanical Properties” describes a series of unsaturated polyamides that are extremely tough, possess highly tunable mechanical properties, and are biocompatible. The manuscript is well written and data support the conclusions. The observation of alkene stereochemistry dictating bulk properties such as tensile strength, modulus, and glass transition of the resulting elastomers is interesting and noteworthy.

I recommend publication after the following changes are made and comments addressed.

Major

A. In particular the paper has no statistical analysis of the data. Please provide statistics and state in the manuscript that the changes observed are significant or not.

We thank the reviewer for pointing this oversight out. Statistical analysis on the cell proliferation data was performed using an ordinary two-way ANOVA test and the following text was added to the manuscript: “Statistical analysis revealed a superior cytocompatibility of C₃A-C₆T *cis* and C₃A-C₈T compared to the PLLA control, while C₃A-PEGT was found to support cell proliferation less than PLLA”.

B. Also a control 8 week in vivo experiment with nylon 6 or nylon 6,6 is required as the paper compares the performance of these new nylon polymers with nylon 6 and nylon 6,6 in all other experiments except for the in vivo one.

We appreciate the comment and point of concern. PLLA is considered a gold standard for implantable synthetic polymers and does not degrade over the time frame of the *in vivo* studies (8 weeks), and hence we selected it as a suitable control for this experiment. In hindsight, it could be considered ideal to have selected a nylon as a control however feel that performing another animal study would be unnecessary and that the sacrifice of additional animal is not ethically justified to provide a control experiment given the broad body of literature that is available on Nylons in biomedical use. Indeed, the biocompatibility of nylons has long been established in a variety of different animal models (Gehrke, S.A.; Aramburú Júnior, J.; Pérez-Díaz, L.; Treichel, T.L.E.; Dedavid, B.A.; De Aza, P.N.; Prados-Frutos, J.C. New Implant Macrogeometry to Improve and Accelerate the Osseointegration: An In Vivo Experimental Study. *Appl. Sci.* **2019**, *9*, 3181; Higgins, D.M.; Basaraba, R.J.; Hohnbaum, A.C.; Lee, E.J.; Grainger, D.W.; Gonzalez-Juarrero, M. Localized Immunosuppressive Environment in the Foreign Body Response to Implanted Biomaterials. *Am J Pathol*, **2009**, *175* (1), 161-170) and even tissue spaces in human subjects (Kim, Su-Min et al. "Comparison of changes in retentive force of three stud attachments for implant overdentures." *The journal of advanced prosthodontics* vol. 7,4 (2015): 303-11. doi:10.4047/jap.2015.7.4.303; Thomas, Daniel J. "3D printing durable patient specific knee implants." *Journal of orthopaedics* vol. 14,1 182-183. 5 Jan. 2017, doi:10.1016/j.jor.2016.12.015; Guthe, Hans Jørgen Timm et al. "Effect of topical anaesthetics on interstitial colloid osmotic pressure in human subcutaneous tissue sampled by wick technique." *PloS one* vol. 7,2 (2012): e31332. doi:10.1371/journal.pone.0031332). While none of these studies can be used as a direct comparator and control for our work, clearly this significant body of evidence shows that nylons are regarded as biocompatible materials thus negating the need for an additional control to ascertain an already established conclusion.

Finally, the Nature publishing group recommends the ARRIVE guidelines for reporting animal research: "Improving Bioscience Research Reporting: The ARRIVE Guidelines for Reporting Animal Research." *PLoS Biol.* *8*, e1000412, (2010). These guidelines suggest encourage authors to thoroughly assess the necessity and context for each animal study. Given our statements above, we believe that it is neither ethical nor necessary to perform an additional animal study in the context of this work.

C. Please perform a cyclic shape memory experiment and provide the results in the manuscript or SI. Cyclic shape memory experiments have been performed, examining differences that could result from stereochemistry as well as the choice of comonomer. In all materials, regardless of composition, a slight decrease in shape memory performance was found, but this is unsurprising for thermoplastic shape memory materials. The greater flexibility that is observed in the materials that contain the ether comonomer (C₃A-C_{DEGT}) are more likely to result in decreased shape recovery as the materials are cyclically deformed, while the analogues closer to traditional nylon structures (C₃A-C₆T) are more robust.

We have added the following text in the shape memory section:

"We also investigated the durability by performing cyclic shape memory experiments, examining differences that could result from stereochemistry as well as choice of thiol comonomer (Supporting Information Figure S30. Table S5). Regardless of composition, a slight decrease in shape memory performance was found, especially after 10 cycles. This diminished cyclic repeatability in the shape memory behavior is not unexpected for thermoplastic shape memory materials, and may also be attributable to fatigue in the macroscopic film. Importantly, potential medical applications that make use of shape memory behavior to produce minimally invasive devices do not require a material to behave as an actuator. Rather, a single shape recovery response is necessary for minimally invasive surgeries. The performance of the polyamides displayed here is more than adequate for such applications."

All Comments

1. Page 1, line 12 – please remove the phrase long-term. Long-term biocompatibility is 1 or 2 years not 8 weeks. For example PCL.

We thank the reviewer for this comment and have changed the wording to avoid the phrase “long-term” when discussing *in vivo* data.

2. Page 4, Table 1. Please add nylon 6 and nylon 6,6 to the table as this is the control polymers used to compare performance of the new polymers, as shown in Figure 2

We have added nylons to Table 1 and included their measured mechanical properties from melt processing thin films from commercial pellets. We have not performed SEC analysis on the samples since they are very insoluble in common solvents (when compared to our polyamides) and thus require special GPC instrumentation/conditions for analysis that is not readily accessible and we do not believe will provide any useful information.

3. Page 5. Please provide statistics for the data collected on the Young's modulus. Are these values different from one another? Similarly for the stress vs strain curves are the data collected at different temperatures different? It appears so – but further analysis is required.

Statistical analysis was performed to determine the differences as a function of stereochemistry as well as composition (with fixed stereochemistry). It was found that there is a statistical difference between the elastic moduli of materials with very high (~75% and higher) *cis* content and those with low *cis* content (~below 50%). However, the elastic moduli differences between the 35% and 46% *cis* materials was found to be statistically insignificant. All other measured mechanical properties were also found to be statistically insignificant with regards to the reported differences.

We have added the following text to the “Thermal and Mechanical Properties” section:

“...It was found that there is a statistical difference between the elastic moduli of materials with very high ($\geq 73\%$) *cis* content and those with low ($\leq 46\%$) *cis* content. However, the elastic moduli differences between the 35% and 46% *cis* materials was found to be statistically insignificant... The strain at break for **P4** ($\epsilon_{\text{break}} = 138 \pm 87\%$) was not drastically different from **P1** and was found to be statistically insignificant...”

Regarding the composition of the thiol comonomer in the polyamide backbone, the elastic moduli of the materials was found to be statistically different with varying lengths. The only exception to this was with the material containing the ether-containing dithiol unit, which was not statistically different from the C₆T unit and the C₃T unit. The strains at failure were also found to be statistically different from compositions, with the exception of the C₁₀T and C_{DEGT} units, which were insignificantly different. Yield stresses also varied significantly among compositions, again with the exception of the C_{DEGT} and C₃T linkers; yield strains were not statistically different; stresses at break were statistically the same between C₆T, C₈T and C₁₀T units, and C₈T was not different from C₁₀T and C_{DEGT} units, as well.

We have amended the text in the “Modulation of Material Properties by Varying Thiol Comonomer” section:

“...Unsurprisingly, as the number of carbon atoms increased in the thiol moiety (C₃, C₆, C₈, C₁₀), both *E* and σ_{stress} decreased (Figure 5A) and these differences were found to be statistically significant. However, as the linker length increased, the ductility of the materials greatly improved (C₃A-C₃T, $\epsilon_{\text{break}} = 24 \pm 3\%$ versus C₃A-C₁₀T, $\epsilon_{\text{break}} = 313 \pm 19\%$) and these were also statistically significant...”

“...In fact, C₃A-C_{DEGT} was more ductile ($\epsilon_{\text{break}} = 318 \pm 70\%$) than other materials and was found to be statistically different from all compositions except for C₃A-C₁₀T. Remarkably, it also possessed the greatest modulus ($E = 1,570 \pm 76$ MPa) and yield stress (78.5 ± 1.7 MPa) for the entire polyamide series, although these values are not statistically different from the C₃T or C₆T polymers...”

4. Page 6: Line 125: from 0 – 280 °C, 10 K min⁻¹ 125 . Please be consistent in measurement units – K or °C.

We have changed K to °C in this and other instances in the text.

5. Page 7: Line 168: A very long sentence – please rewrite as this is an important concept for the reader. Also please do not use “this” as a subject and noun to start a sentence.

We agree that the sentence was unclear as it was previously written. The single sentence was restructured into two separate sentences:

“The slower recovery of the low *cis* samples may be due to better polymer chain packing that is afforded by relative abundance of hydrogen bonding around individual chains. On the other hand, high *cis* polymers may have a decreased relative concentration of hydrogen bonding because of a more open molecular structure, and therefore reduced interactions with surrounding carbonyl moieties.”

6. Page 7: line 171: Please perform a cyclic shape memory experiment and provide the results in the manuscript or SI. This is an important performance characteristic of the material and needs to be known. Please see response to comment C above

7. Page 8: Line 200: Please justify the choice of a particular cell line: Mouse osteoblasts (MC3T3-E1). Typical biocompatibility studies use NIH 3T3 cells.

Originally, we envisaged that these materials would be suitable for bone tissue engineering, hence a pre-osteoblast line was selected to test cytocompatibility however, we note that this cell line is widely used and generally considered as a suitable cell model for studying cytocompatibility. The following sentence has been added to the manuscript:

“This cell line is generally considered as a suitable cell model for studying material cytocompatibility (Biomaterials, 1993, 14, 263-269; ACS Appl. Mater. Interfaces, 2012, 4, 4966–4975, J Mater Sci: Mater Med, 2015, 26, 116)”.

8. Page 9: Figure 4a and Line 203: excellent cytocompatibility – what does the word excellent refer to in this statement? The results were similar to controls. Or is this not correct? Why is there increased metabolic activity with the polymer present or are the results similar to the control. Please provide some statistical analysis.

“Excellent cytocompatibility” was replaced with “good cytocompatibility”. Statistical analysis on the cell proliferation data was performed using an ordinary two-way ANOVA test and the following text was added to the manuscript: “Statistical analysis revealed a superior cytocompatibility of C₃A-C₆T *cis* and C₃A-C₈T compared to the PLLA control, while C₃A-PEGT was found to support cell proliferation less than PLLA”.

9. Page 9: line 212: PLLA is a good control for biocompatibility but it is not a nylon structure. Please repeat the experiment with nylon 6 or nylon 6,6 and provide the data in the manuscript. Please see response to comment B above

10. Page 11: Line 253: Please check the sentence structure and fix grammatical error: may more processable materials

Thank you for highlighting this mistake. We have changed the text to “... may present a path to processable materials with high toughness.”

11. Page 13 Line 334: Please include page numbers for the ref 18. Surgery 22, (2011) ?????.
We have corrected Ref 18 and it now appears as:

Khadka, A. et al. Evaluation of hybrid porous biomimetic nano-hydroxyapatite/polyamide 6 and bone marrow-derived stem cell construct in repair of calvarial critical size defect. Journal of Craniofacial Surgery 22, **1852-1858**, (2011).

Supplemental Information document:

1. Line 5: ±- superscript

We have amended this formatting issue.

2. Lines 4 and 5: sometimes the sign is before comma and sometimes it's after. Please fix. The following

sequence “, §”

We have amended this formatting issue as well.

3. What's the gender of mice (n = 6) studied?

We used rats in this study, not mice. The rats selected for the animal studies were all male. This information is present in the supplementary information of the paper, in the General methods and materials section under “*In vivo* animal studies” on page S6.

4. Starting from line 235, adjust titles of spectra and maybe fit two spectra for each compound and their titles into one page.

Thanks for this comment. We have adjusted the formatting to reflect this.

5. Line 455: is there any protocol for animal work that you may have followed that you should reference?

The *in vivo* studies were designed with the help of an *in vivo* specialist, taking into account our experimental goals while complying with the 3Rs (replacement, reduction, refinement) rules for animal studies. As such, the protocol developed was not based on any previous publications but can be found in full in the supplementary information under the section “*in vivo* animal studies”.

Reviewer #3 (Remarks to the Author):

The manuscript by Worch et al. entitled “Biocompatible and Elastomeric Polyamides with Stereochemically-Tunable Mechanical Properties” presents the synthesis of a new polyamide series that offers interesting mechanical, shape memory, and *in vivo* compatibility properties for potential use as a biomaterial. The manuscript is generally well written and easy to read. This reviewer suggests major revisions for this manuscript.

Major points:

1) How and why were monomers chosen for this study? What is the rationale for the number of methylenes (linker length, etc.) for these initial structures that were studied? While certainly the stereochemistry plays a role, is there a point where that does not matter (when the monomer lengths are sufficiently) increased? The authors do examine in lesser detail derivatives of the thiol monomers (of different numbers of methylenes and an ethylene oxide derivative) but did not examine derivatives of the alkyne – which is important to understand how further alterations in the chemical properties may enhance, dilute/diminish the stereochemical effects. This also can alter Tg and crystallinity as the methylene number in nylons on both sides of the amides are very important to physical properties (particularly noncovalent interactions between the polymer structures).

These are all very good questions that we are happy to address. We will try and treat these point by point below for clarity. Before doing so, we would like to point out that we have amended some of the text in the “synthesis of polyamides” section to include the following sentences. We believe this will shed light on these issues that are highlighted:

“Early in our investigations, we synthesized additional propiolamide monomers with longer aliphatic chains between the nitrogen atoms (such as C₄A, C₅A, C₆A and C₇A) and screened them in the step-growth polyaddition reaction (data not shown). All monomers with longer methylene spacers were reactive with 1,6-hexanedithiol, however the resultant polymers displayed limited solubility using our reaction conditions (see Table 1 for experimental details). These polymers would commonly precipitate from the reaction mixture within seconds to minutes, even when the thiol co-monomer was varied, to afford an intractable, powdery material that was presumably low molecular weight. Since we could not adequately characterize these polyamides due to their limited processability, we excluded them from this study. It is possible that other reaction conditions (e.g. changing the solvent to hexafluoroisopropanol (HFIP)) could help solubilize these polymers and afford higher molecular weight materials. Although we did not use HFIP, we did screen other polar-aprotic solvents throughout our study, such as hexamethylphosphoramide (HMPA), but we found DMSO to be the best solvent overall and critical in promoting solubility.”

Q: How and why were the monomers chosen for this study? What is the rationale for the number of methylenes (linker length, etc.) for these initial structures that were studied?

A: 1) The propiolamide monomer synthesis only proceeds efficiently in water and thus this limits the choice of suitable diamines. When organic co-solvents such as THF or DMF were employed to help solubilize reactants, we saw Michael addition-type reaction products of the amine to the propiolate. This meant that more hydrophobic amines, such as aromatic or larger aliphatic (> 7 methylene units) diamines, did not afford propiolamide products in suitable yields.

2) For this study we chose the primary propiolamide to have 3 methylene moieties (C₃A) and the thiol to have 6 methylene units (1,6-hexanedithiol) since this polymer exhibited the best solubility over the entire range of reaction conditions that were necessary to control stereochemistry. Polymers formed from propiolamides with 4 (C₄A), 5(C₅A), 6(C₆A), and 7(C₇A) methylene spacers were comparatively less soluble, especially C₆A. In fact, C₆A polymers were insoluble and precipitated out of solution when co-polymerized with any thiol comonomer.

Q: While certainly the stereochemistry plays a role, is there a point where that does not matter (when the monomer lengths are sufficiently) increased?

A: It is possible that the stereochemical effects may be diminished according to the size of the co-monomers (propiolamides and/or thiols) and there could certainly be a point where the effects are negligible. We did not investigate stereochemical effects in-depth for C₃A-C_xT polymers since the reaction condition variations to alter stereochemistry were optimized for the C₃A-C₆T polymer. However, we plan to investigate stereochemical effects in these polymers in future studies, especially the derivative with ether linkages.

Q: The authors do examine in lesser detail derivatives of the thiol monomers (of different numbers of methylenes and an ethylene oxide derivative) but did not examine derivatives of the alkyne – which is important to understand how further alterations in the chemical properties may enhance, dilute/diminish the stereochemical effects.

A: As discussed partly in the first question above, we did screen other alkyne (propiolamide) monomers in our initial studies (C₄₋₇A), but we did not include them for further analysis in this manuscript because these monomers afforded polymers that were very insoluble compared to polymers obtained from C₃A. Also, as the size (# of methylene units) of the diamine precursor increased, the yield and purity of the alkyne monomer was greatly diminished when synthesizing the propiolamides. All of these factors ultimately led us to investigate the C₃A derivative in numerous polymers by varying the stereochemistry and/or thiol comonomer. We were able produce other polymers using different alkyne monomers, but their insolubility precluded their full characterization and likely tempered resultant molecular weights – we generally found them to be powdery solids that did not form good films.

Q: This (linker lengths) also can alter Tg and crystallinity as the methylene number in nylons on both sides of the amides are very important to physical properties (particularly noncovalent interactions between the polymer structures).

A: We agree that this can be very important for influencing the thermal and mechanical behaviour of polyamides. We attempted to synthesize many other polymers using different alkyne monomers. At times, we did observe partial solubility under these reaction conditions for other polymers (e.g. C₇A-C_xT), but they were not reproducible enough to analyze adequately.

2) At the very least, it is important to include a synthetic analog to nylon 6,6 (Nylon 6 and 6,6 are used in Figure 2 but nylon 6, 6 would be needed if a 6, 6 analog is made). It would also be important to have a better a nylon control (3,14) that is more analogous to the lengths between the amides of the synthetic analog that is studied in details. Also, the authors attribute the shape memory properties to the H-bonding and again, it is important to have more accurate analogous nylons to compare the physical/shape memory properties.

We attempted to synthesize an analogue to nylon 6,6 by polymerizing C₆A alkynes with various thiols (although the thiol co-monomer would have to contain zero methylene units to truly be analogous to nylon 6,6). As previously mentioned, polymers made using the 6 methylene alkyne derivative immediately

precipitated from solution to yield powdery solids (of presumably low molecule weight) that were insoluble in common organic solvents.

We did synthesize a Nylon 3,14 derivative from melt-polycondensation of 1,3-diaminopropane and tetradecanedicarboxylic acid. The resultant polymer was not high enough molecular weight to make a suitable film; however, it had a physical appearance similar to Nylon 6 and Nylon 6,6. Investigation of the thermal properties using DSC revealed that it was semi-crystalline (dissimilar to our C₃A-C₆T polyamides) as evidenced by a defined melt and crystallization events (Figure S27). Nylon 3,14 did not appear to exhibit any shape memory behaviour, although we mention this with caution since the films were too brittle to investigate fully. Nylon 6 and Nylon 6,6 do not (and did not in our hands) exhibit shape memory behaviour.

3) It is not understood why all of the chemical analogs at the end of the manuscript (and their subsequent physical properties) were not included in the original Table 1. It would be more useful to have all of the analogs displayed side-by-side to further gain insight into the role of linker length. Why was the C3A-C6T analog selected for the extensive studies and not the ethylene oxide derivative? This was not explained in the manuscript.

We appreciate the reviewer's comment concerning Table 1, however we believe that the large table with full characterization data for every polymer (Table S6) is better suited for the Supporting Information since its inclusion in the manuscript would disrupt its flow. However, we have included a separate table in the manuscript that describes the properties of the polyamides with different composition in the same manner as Table 1. This is referred to as Table 3 and can be found just before the conclusion section.

4) What is the mechanism of shape memory? Discerning this is central to the impact of this manuscript. How/why does the ethylene oxide derivative exhibit shape memory?

Shape memory in polymers is a multifaceted phenomena that commonly is only attributed to intermolecular interactions, although this is not solely the case. Chain entanglements and entropic free energy will dictate the permanent and temporary network configurations in the same manner as hydrogen bonding. While the polyamides presented here all have the propensity for hydrogen bonding-related shape memory effects, the reviewer raises an interesting point about the presence of the ether-containing composition as still having shape memory. While poly(ethylene oxide) would have substantially increased mobility that could reduce shape fixation, the segments utilized here contain only two ether bonds per repeat unit (diethylene glycol) and would therefore, we believe, not increase flexibility enough to transition the shape memory response away from being dominated by the hydrogen bonding. An important note to make here however, is that these thermoplastic polymers display good initial shape recovery, on par with crosslinked materials with permanent netpoints that help maintain the original shape. This behaviour is attributable to both the hydrogen bonding as well as the aforementioned entanglements and entropic free energy of the polymer.

We have added the following text to the "Shape Memory Properties" section:

"...An important note to make is that these thermoplastics display good initial shape recovery, on par with crosslinked materials containing permanent net-points which help promote their original shape structures. In our system, it is likely that these features are a result of robust chain entanglements (due to high molecular weight) and entropic free energy contributions combined with a high degree of hydrogen bonding among the amide moieties..."

5) How is the bulk structure of the materials impacted by stereochemistry? Scattering experiments could be completed to determine this as this could give insight into the mechanism of shape memory.

We agree that this is a very important consideration and we are currently investigating stereochemical effects on the bulk structure in more detail. DSC (Figures S22, 24, 26), DMA (Figure S28) and AFM (Figure S39) analyses of the processed films indicate that the material is amorphous regardless of stereochemistry (and/or composition). However, we are currently probing the structure using WAXS/SAXS and solid-state NMR techniques to better elucidate the stereo-chemical effects in a follow-up study.

6) One cycle of shape memory is shown for each derivative. How many cycles can these materials handle before fatigue?

We thank the reviewer for this comment but note that the definition of fatigue meant by them is not clear. To attempt to address this comment, the role of stereochemistry and composition was briefly examined over 60 cycles with shape recovery/deformation at $T_g + 20$ °C. It was found that the materials rapidly display a slight loss in strain recovery within the initial 10 cycles, after which the strain recovery seems relatively stable for the remainder of the cycling. This is less of a fatigue behaviour and more similar to a settling response.

We have added the following text in the shape memory section:

“We also investigated the durability by performing cyclic shape memory experiments, examining differences that could result from stereochemistry as well as choice of thiol comonomer (Supporting Information Figure S30. Table S5). Regardless of composition, a slight decrease in shape memory performance was found, especially after 10 cycles. This diminished cyclic repeatability in the shape memory behavior is not unexpected for thermoplastic shape memory materials, and may also be attributable to fatigue in the macroscopic film. Importantly, potential medical applications that make use of shape memory behavior to produce minimally invasive devices do not require a material to behave as an actuator. Rather, a single shape recovery response is necessary for minimally invasive surgeries. The performance of the polyamides displayed here is more than adequate for such applications.”

Minor points:

7) The chemical structures in Figure 1 are a bit small, the images should be increased in size.

Thank you for this feedback. We have increased the size of the scheme in Figure 1 and added the SEC chromatograms next to the representative ^1H NMR spectrum.

8) It would be useful to add the T_g data to Table 1

We believe this is helpful and have added the glass transition as determined by DSC analysis in Table 1. This table also now includes data for Nylon 6 and Nylon 6,6.

9) It is important to have more discussion about the importance of the modulus (and comparison to other known materials) such that the general readership of Nature Communications can understand the relevance of these data compared to common materials. (Literature)

In general, we have expanded the introduction to contextualize polyamide biomaterials and how this material specifically compares to those existing formulations:

“Polyamides have been a biomaterial of choice for decades and they have been extensively developed for applications ranging from use as sutures^{9,10} or membranes¹¹ to vascular applications¹² because of their toughness, low-cost and outstanding biocompatibility.⁸ Polyamides are also widely used in bone engineering as a consequence of the materials' high strength and flexibility which is due to the extensive degrees of both crystallinity and hydrogen bonding.¹³⁻²¹ Even though this may be useful in selected orthopedic or vascular applications, these types of materials are typically difficult to process and functionalize, which has ultimately limited their performance in other applications. An ideal durable biomaterial platform would incorporate the thermomechanical and biological performance of polyamides while displaying enhanced processability and advanced functionality, such as shape memory for minimally invasive device designs.”

To address the properties of this material compared to other biomaterials we have added the following sentence in the “Thermal and Mechanical Properties” section:

“...Overall, the polyamides possessed high moduli with moderate ductility. Specifically, the mechanical properties are quite comparable to Nylon 6,6 which is a useful biomaterial for numerous applications...”

We have also added this sentence to the conclusion:

“...Our polyamides have comparable mechanical performance to nylons, but are strikingly more adaptive and easier to process...”

10) PLLA was used as a control for the in vivo studies. Another control should likely be analogous nylons. Consistent controls should be used for the studies throughout this work. (Literature)
Please see our response to Point B, reviewer 2.

11) There are a few typos in the paper that should be fixed.
Thank you for reading the manuscript carefully and bringing this to our attention. We have found several minor typos and corrected them.

Reviewers' comments:

Reviewer #1 (Remarks to the Author):

I feel that the points which have raised in the previous round of review have now been satisfactorily addressed by the authors, so the manuscript can be published now.

Reviewer #2 (Remarks to the Author):

The authors addressed all of the comments and the manuscript is improved. However the responses to the comments regarding the in vivo biocompatibility and in vitro cytocompatibility are less than satisfactory. These comments were suggested to improve the quality and potential impact of the paper.

1. With regards to the in vivo biocompatibility, I agree with the ARRIVE guidelines and the statement "While none of these studies can be used as a direct comparator and control for our work, clearly this significant body of evidence shows that nylons are regarded as biocompatible materials" The fact is the in vivo experiment was not well-planned. If one synthesizes new nylons they should be compared to conventional nylons as all the other experiments in the manuscript do. It would be like doing an experiment to taste three apples and using an orange as a control instead of an autumn crisp. I do appreciate that this would be a significant undertaking in the lab, and therefore will acknowledge that the control of PLLA is acceptable, but disappointing. Why did you chose to answer the comment but not include the information in the revised manuscript on pages 9-11.

Please add a sentence and the references in your response to the manuscript concerning the biocompatibility of nylons.

2. Also please do not use the word "virgin" to describe a sample type.

3. I do ask that the authors perform the cell toxicity studies with NIH3T3 cells. This experiment will take one week. I agree that other authors have tested toxicity of their materials against MC3T3-E1, in fact I can find papers for almost any cell line for any material. NIH3T3 cells are suitable, a well-accepted cell line used by a very very large number of research and industrial labs, and a cell line listed in regulatory documents for testing of polymers to be used in medical devices.

Reviewers' comments:

Reviewer #1 (Remarks to the Author):

I feel that the points which have raised in the previous round of review have now been satisfactorily addressed by the authors, so the manuscript can be published now.

We thank the reviewer for providing valuable feedback that improved our manuscript.

Reviewer #2 (Remarks to the Author):

The authors addressed all of the comments and the manuscript is improved. However the responses to the comments regarding the in vivo biocompatibility and in vitro cytocompatibility are less than satisfactory. These comments were suggested to improve the quality and potential impact of the paper.

1. With regards to the in vivo biocompatibility, I agree with the ARRIVE guidelines and the statement “While none of these studies can be used as a direct comparator and control for our work, clearly this significant body of evidence shows that nylons are regarded as biocompatible materials” The fact is the in vivo experiment was not well-planned. If one synthesizes new nylons they should be compared to conventional nylons as all the other experiments in the manuscript do. It would be like doing an experiment to taste three apples and using an orange as a control instead of an autumn crisp. I do appreciate that this would be a significant undertaking in the lab, and therefore will acknowledge that the control of PLLA is acceptable, but disappointing. Why did you chose to answer the comment but not include the information in the revised manuscript on pages 9-11.

Please add a sentence and the references in your response to the manuscript concerning the biocompatibility of nylons.

We agree that these references are important to include and the following sentence has now been added on page 10: “PLLA is considered a gold standard for implantable synthetic polymers and does not degrade over the time frame of the in vivo studies (8 weeks), and hence we selected it as a suitable control for this experiment. Moreover, the biocompatibility of nylons has long been established in a variety of different animal models”. Relevant references (73-74) have now been added to the text.

2. Also please do not use the word “virgin” to describe a sample type.

The sentence “virgin samples” has been reworded to “samples that have not been implanted”.

3. I do ask that the authors perform the cell toxicity studies with NIH3T3 cells. This experiment will take one week. I agree that other authors have tested toxicity of their materials against MC3T3-E1, in fact I can find papers for almost any cell line for any material. NIH3T3 cells are suitable, a well-accepted cell line used by a very very large number of research and industrial labs, and a cell line listed in regulatory documents for testing of polymers to be used in medical devices.

While in normal times, it may be easiest to simply comply with the reviewers' request, the coronavirus pandemic is currently preventing us from undertaking experimental work and hence at the present time, complying with this request is, unfortunately, not possible. We, however, do not believe that these experiments are strictly required anyway and are happy to explain this position below.

MC3T3 cells are also extensively used and reported for proliferation assays in materials science, both in 2D and 3D cell studies. Indeed, in our view, these are very widely used for the same purposes reported in this manuscript, namely, proliferation assays (to reassure us of the low cytotoxicity of the materials prior to *in vivo* study). The “3T3” designation refers to the abbreviation of “3-day transfer, inoculum 3×10^5 cells” which refers to the so-called “3T3 protocol”, used to generate spontaneously transformed (immortalized) cell lines with a stable growth rate (*Jpn. J. Oral Biol.*, 23: 899-901, 1981). Both the MC3T3-E1 cells we used and the NIH3T3 cell line noted by the reviewer are derived using the same protocol, are immortalised cell lines, and have a fibroblast-like morphology. The only difference between the lines is that NIH3T3 are derived from the embryo tissue of a Swiss albino mouse whereas MC3T3 are derived from the calvaria (skull) of C57BL/6 mouse.

We argue that, given the purpose of our cell study was to demonstrate cytocompatibility, neither the mouse breed nor the tissue of origin is of any relevance in this study and hence it is reasonable to conclude that there will be little to no outcome differences observed (*i.e.* the materials are not cytotoxic) between the cells used for proliferation and the ones proposed by the reviewer. Therefore, we believe the requested additional cell study will not add a meaningful contribution to the manuscript and would be redundant.

We note that the sentence “This cell line is generally considered as a suitable cell model for studying material cytocompatibility” and relevant references (70-72) were added in response to this reviewer’s comment in the last set of reviews to address these concerns.